# Dispersible hydrogel force sensors reveal patterns of solid mechanical stress in multicellular spheroid cultures

Wontae Lee [1], Nikita Kalashnikov [1], Stephanie Mok [1], Ruba Halaoui[2,3], Elena Kuzmin [2,4], Andrew J. Putnam [5], Shuichi Takayama [5,6], Morag Park[2,4], Luke McCaffrey[2,3], Ruogang Zhao [7], Richard L. Leask[1,8,9] & Christopher Moraes [1,2,9]

Understanding how forces orchestrate tissue formation requires technologies to map internal tissue stress at cellular length scales. Here, we develop ultrasoft mechanosensors that visibly deform under less than 10 Pascals of cell-generated stress. By incorporating these mechanosensors into multicellular spheroids, we capture the patterns of internal stress that arise during spheroid formation. We experimentally demonstrate the spontaneous generation of a tensional 'skin', only a few cell layers thick, at the spheroid surface, which correlates with activation of mechanobiological signalling pathways, and balances a compressive stress profile within the tissue. These stresses develop through cell-driven mechanical compaction at the tissue periphery, and suggest that the tissue formation process plays a critically important role in specifying mechanobiological function. The broad applicability of this technique should ultimately provide a quantitative basis to design tissues that leverage the mechanical activity of constituent cells to evolve towards a desired form and function.

[1] Department of Chemical Engineering, McGill University, Montréal H3A 0C5 QC, Canada. [2] Rosalind and Morris Goodman Cancer Research Centre, McGill University, Montréal H3A 1A3 QC, Canada. [3] Department of Oncology, McGill University, Montréal H4A 3T2 QC, Canada. [4] Department of Biochemistry, McGill University, Montréal H3G 1Y6 QC, Canada. [5] Department of Biomedical Engineering, University of Michigan, Ann Arbor 48109 MI, USA. [6] Department of Biomedical Engineering, Georgia Institute of Technology and Emory University, Atlanta 30332 GA, USA. [7] Department of Biomedical Engineering, State University of New York at Buffalo, Buffalo 14260 NY, USA. [8] Montreal Heart Institute, Montréal H1T 1C8 QC, Canada. [9] Department of Biological and Biomedical Engineering, McGill University, Montréal H3A 2B4 QC, Canada. Correspondence and requests for materials should be addressed to C.M. (email: chris.moraes@mcgill.ca)

Within 3D tissues, cells actively exert highly localized forces on their surroundings to drive morphogenetic remodeling of tissue shape and structure, ultimately directing tissue function[1-3]. The generation and transmission of force plays a critical role in biological processes, from organ development[4-7], to maintaining tissue homeostasis[8], to driving disease progression[9-11]. The ability to measure localized forces within multicellular tissues is vital to understand developmental processes and may be of practical utility in developing novel tissue engineering strategies. However, measuring cellular-scale and multi-directional forces within living, three-dimensional tissues remains challenging.

Recent advances in imaging technologies[12,13], engineered genetic sensors[14-16], and microfabricated tissue models[17-20] have critical limitations in estimating cell-generated forces at various length scales. Anchored microfabricated cantilevers can serve as tension gauges to determine tissue-scale forces, but are limited to measurements at the tissue periphery, and reflect forces generated by the entire tissue[17,21-25]. Förster resonance energy transfer (FRET)-based molecular tension probes can readout cellular force with sub-cellular spatial resolution, but measurements at this scale cannot be easily integrated to determine cellular stresses that drive tissue remodeling. To measure forces at supra-cellular length scales, traction force microscopy-based methods can be applied for single cells[26-29] and cell colonies[30,31]. This approach requires observing the deformations caused by the mechanical activity of a cell, and calculating stresses based on knowledge of tissue stiffness[32]. However, this is particularly challenging for large tissue deformations, or when tissue rigidity is altered during remodeling[33]. Furthermore, tissue rigidity is non-linear for natural biopolymer extracellular matrices[34,35], and may change drastically for even small deformations of polymer fiber networks[36,37]. Hence, traction force microscopy cannot be readily applied when studying highly dynamic tissues.

To apply traction-based strategies more broadly, small domains of precisely defined mechanical properties can be created within mechanically complex tissues. Campàs et al. injected incompressible oil microdroplets into 3D cell aggregates and live embryonic tissue, and measured their shape deformation to calculate local anisotropic forces present during remodeling[38]. The oil droplets deform sufficiently to measure kilopascal-scale stresses, but their incompressibility does not allow isotropic deformation. Hence, isotropic stress components are neglected, and the absolute magnitude of local stress cannot be measured. Using compressible materials such as hydrogels could circumvent this issue, and Dolega et al. examined the isotropic compaction of stiff ($E$ ~15 kPa) hydrogel microdroplets embedded in multi-cellular tissues when the tissues were loaded externally[39,40], demonstrating that external stresses do not propagate uniformly through tissues. While useful for mapping variations in tissue rigidity, these stiff hydrogel microdroplets are unable to measure forces generated by cells, which can be 3 orders of magnitude smaller than the external loads applied in their study. More recently, others have reported softer alginate sensors ($E$ ~1.5 kPa) that can detect cell-generated forces[41], but these can only be applied to distinguish large compressive stress (100 s of Pa), and were applied primarily to measure tightly localized subcellular spatial variations in stress between cells. Furthermore, the well-established viscoelastic non-linear behavior of calcium-crosslinked alginate, and sensitivity of this material to external calcium fluxes requires careful interpretation of results, and limits broad utility of the system.

Building upon these strategies, we present a technique to quantitatively measure cell-generated mechanical stresses within 3D tissues at the cellular length scale, by developing a polyacrylamide hydrogel formulation that is (1) sufficiently soft ($E$ ~0.15 kPa) so as to deform under a few Pascals of cell-generated stresses; (2) compressible, linearly elastic, and mechanically stable over a broad range of strains and tissue culture conditions; and (3) can be fabricated into microspheres to be incorporated into engineered tissues (Fig. 1a, b). The low stiffness of these polyacrylamide microspherical stress gauges (MSGs) was achieved by incorporating chain-terminating fluorescent monomers into high polymer-content hydrogel formulations, resulting in microscale, compressible structures with well-defined mechanical properties that can be imaged within the tissue. Observed deformations are readily converted to obtain absolute, directional, and local tissue stresses in situ.

As a first application of this technology, we disperse MSGs into 3D multicellular spheroid (MCS) cultures, and quantitatively map highly localized and symmetric radial and circumferential stresses within the tissue. We provide experimental evidence supporting previous inferences[42] and computational predictions[43] that a 'skin' of tension forms around multicellular spheroids. Furthermore, we demonstrate that this tensional skin can be only a few cell layers thick, forms rapidly in engineered tissues, and correlates well with markers of known mechanobiological activity. We then use a combination of imaging and computational modeling techniques to determine how these stress patterns might arise within the MCS, demonstrating that spheroid formation dynamics regulate internal mechanical stresses in this widely used culture model.

## Results

**MSG material design and fabrication.** To fabricate the MSGs, we required a biomaterial that would incorporate well in long-term cell cultures, display compressible, stable, and linear elastic material properties over the range of expected stress measurements, and could be imaged to measure sensor deformation. Polyacrylamide hydrogels were selected for their biocompatibility, non-degradability in culture, compressibility, ease of surface functionalization, and historical use in traction force microscopy[25]. Most importantly, polyacrylamide stiffness can be manipulated over a large range by increasing or decreasing polymer content.

Microdroplets of stiff polyacrylamide have been produced in an immiscible two-phase kerosene/aqueous system, either via stirred emulsion[44] or microfluidic droplet generation[45]. Applying these techniques to produce soft polyacrylamide MSGs was initially unsuccessful, because small volumes of soft polyacrylamide formulations with low polymer content are more susceptible to quenching of the free radical polymerization reaction via solubilized oxygen. Displacing oxygen with excessive bubbled nitrogen was not sufficient to consistently produce the desired low-stiffness MSGs, but did enable gelation of polyacrylamide formulations containing higher/stiffer polymer content.

To reduce the stiffness of high-content formulations, a chain-terminating fluorescent monomer was incorporated into the polymer backbone[46], to partially disrupt the polymer network (Supplementary Figure 1a–c). We experimentally determined via shear rheometry that adding fluorescein o-methacrylate during polymerization can reduce the shear modulus of the hydrogel network by five-fold, while maintaining linear elastic properties over a large deformation range (Supplementary Figure 1d). This serves the dual purpose of persistently labeling the hydrogel material for fluorescent imaging (Fig. 1d). To ensure compatibility with our intended application, we confirmed that hydrogels labeled with the selected fluorescent chain-terminating monomer retain their fluorescence through all sample handling steps (Supplementary Figure 2), and do not alter cell viability (Supplementary Figure 3).

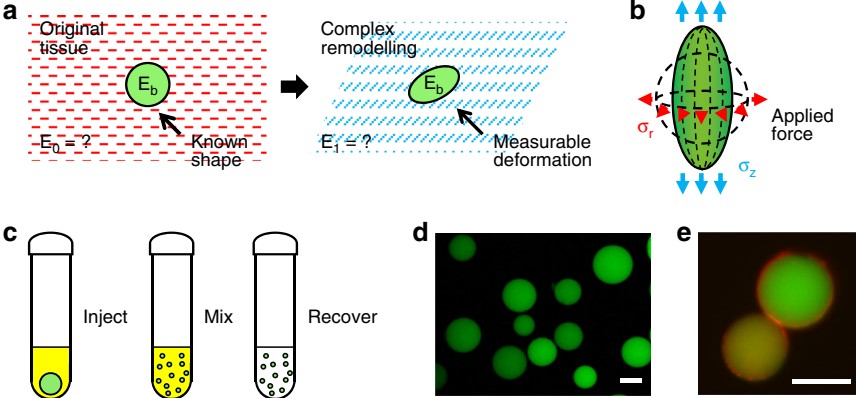

**Fig. 1** Conceptual overview of the microspherical stress gauge (MSG) system. Soft ($E \sim 0.15$ kPa), compressible, and fluorescently labeled hydrogel microspheres can act as sensors of cell-generated mechanical stress. **a** Changes in the shape of MSGs embedded within model tissues can be used to determine the local cell- and tissue-generated mechanical stresses that arise during complex remodeling, such as during morphogenesis, development, or disease progression. In contrast with conventional traction force microscopy techniques, this approach can be applied to study forces in systems with dynamic tissue mechanical properties, which commonly occur during remodeling processes. **b** Measuring deformation of the hydrogel microspheres is sufficient to calculate the local isotropic and anisotropic stress components in the surrounding remodeling tissue. In a spheroid culture system, MSGs are expected to deform in a symmetry-simplified combination of axial and radial stresses. **c** To fabricate the hydrogel MSGs, aqueous polyacrylamide components and fluorescein methacrylate monomers were mechanically dispersed in an immiscible kerosene phase, and allowed to polymerize, producing **d** fluorescently labeled polydisperse MSGs (green; scale bar = 50 µm). **e** Functionalization of the MSG surface with collagen I (red) produces an extracellular matrix coating that is limited to the surface of polyacrylamide microspheres (confocal section of different sized beads shown), that facilitates incorporation of the MSGs into engineered tissues (scale bar = 25 µm)

We tested stirred emulsion and microfluidic droplet generation methods to fabricate MSGs. Stirred emulsions (Fig. 1c) readily produced polydisperse MSGs (Fig. 1d, e; Supplementary Figure 4b; average diameter ~50 µm), which could be sorted by sequential centrifugation steps (Supplementary Figure 5). Microfluidic droplet generation produced larger monodisperse MSGs with tunable diameters (Supplementary Figure 6), but the large swelling ratios of these soft polymer formulations (Supplementary Figure 7) made it difficult to produce monodisperse soft MSGs smaller than 100 µm. Hence, stirred emulsion MSGs (3% acrylamide/0.06% bisacrylamide with 100 µg/mL fluorescein o-methacrylate) were used in this work, but the inherent MSG polydispersity required careful measurement of MSG size at the zero-stress state.

**MSG mechanical characterization and analysis of deformation**. The selected polyacrylamide formulations exhibited isotropic linear elastic behavior over a large strain range (up to 80%) and frequencies (0.001–1 Hz), as confirmed by shear rheometry of bulk samples (Fig. 2a; Supplementary Figure 8). Although others have reported that standard polyacrylamide exhibits non-linear stiffening at high strains[47], we did not observe any appreciable loss modulus or non-linear stiffening behavior in these softer formulations. To characterize the Poisson's ratio, hydrogel strings, ~1 mm in diameter, were fabricated and mechanically stretched (Fig. 2b, Supplementary Figure 9). Consistent with literature reports[48], the ratio of transverse strain to axial strain was constant at $\nu = 0.3$ up to strains of 120%. These values establish the MSG strain limits for accurate measurements when using this technique (Fig. 2b).

Since the mechanical properties of the hydrogel may change when polymerized in microdroplet emulsion form, we applied an osmotic pressure measurement technique[39] to characterize the material properties of MSGs directly (see Supplementary Methods). Briefly, a long-chain dextran solution was used to exert isotropic osmotic pressures on MSGs, and deformation of the MSG was calibrated against osmotic deformation of a bulk sample with known mechanical properties. Using this technique,

the apparent shear modulus of the MSGs was determined to be $60 \pm 3.5$ Pa (Fig. 2c; Supplementary Figure 10). The average modulus is slightly higher than rheometry-based measurement of bulk polyacrylamide modulus, likely due to the low-oxygen environment in which MSGs are polymerized, which is well known to affect polymerization kinetics and may thus affect the hydrogel mechanical properties. This directly measured value of apparent MSG shear modulus was used in all subsequent simulations and analyses.

To incorporate MSGs into tissues, MSGs were functionalized with extracellular matrix molecules using standard Sulfo-SANPAH-based crosslinking chemistry. We confirmed that MSGs can be coated with collagen to ensure incorporation into the tissue of interest (Fig. 1e), and verified experimentally that the collagen coating does not significantly alter MSG mechanical properties (Supplementary Figure 10e). To test whether MSG mechanics were altered within multicellular spheroid tissue cultures, we cultured MSGs in spheroids and released them by detergent-based lysis. The mechanical properties of the MSGs remained unchanged (Supplementary Figure 10f), consistent with observations that collagen does not penetrate into the MSG (Fig. 1e). Hence, mechanical properties of the MSGs can be considered constant even after embedding in the tissue of interest.

The characterized mechanical properties were used to develop the finite element models necessary to relate observed deformation with multi-directional stress magnitudes. Although a full 3D finite element analysis for individual MSGs is possible, we took advantage of the symmetry inherent in our intended application to simplify the models required. Hence, we assumed uniform tensile and compressive loads along the MSG axial (z-direction), and radial (r-direction) axes (Fig. 2d), and applied a full parametric sweep of axial and radial strains to determine the unique combination of external loads needed to create each observed MSG shape (Fig. 2e, f; Supplementary Figure 11). As expected, material compressibility is required to decouple the contributions of multi-directional loads, which would not be possible in incompressible oil microdroplets (Supplementary Figure 12).

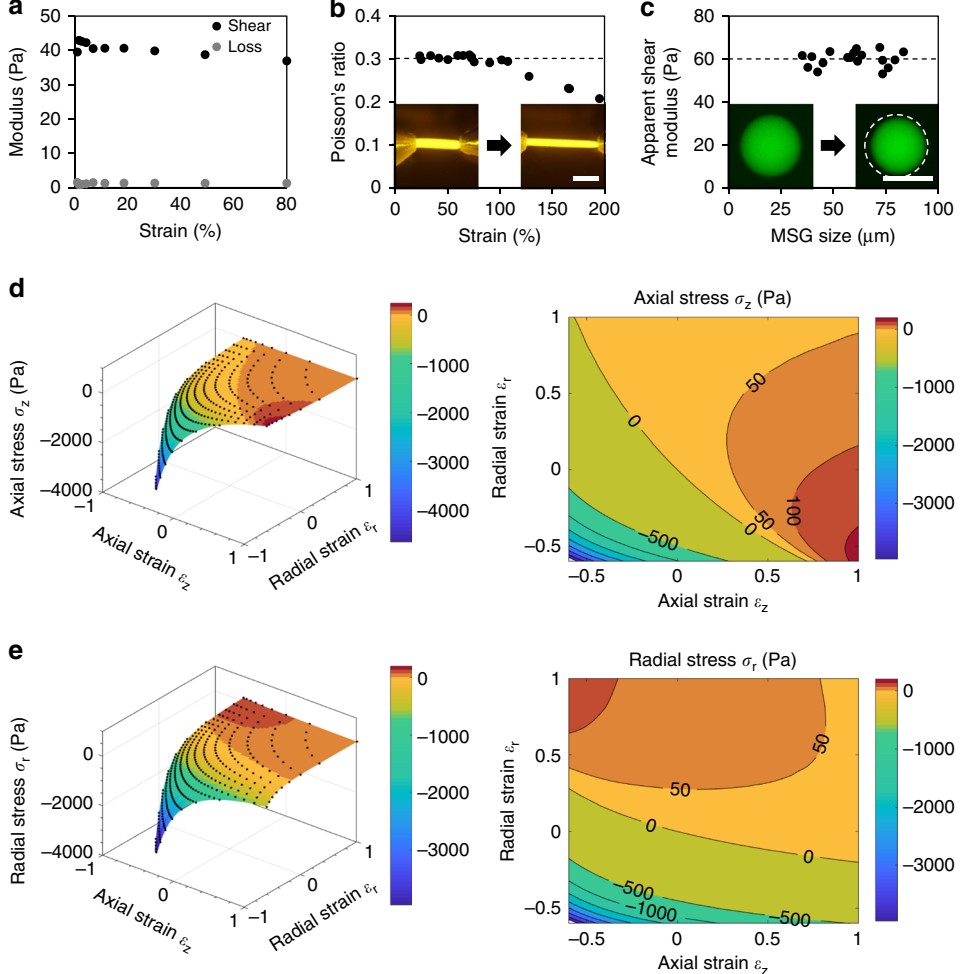

**Fig. 2** Calculating local tissue stresses from observable MSG deformations. **a** Shear rheometry on bulk, fluorescently-labeled polyacrylamide hydrogels indicates linear elastic deformations over large strains, with negligible loss modulus. **b** Deformation of fluorescently labeled polyacrylamide hydrogel strings were used to determine that the Poisson's ratio of the material was constant ($\nu = 0.3$) over the working range of expected deformations (scale bar = 5 mm; $n = 7$). **c** To characterize the mechanical properties of individual MSGs, their deformations were observed under an applied osmotic pressure of 67 Pa. The apparent shear modulus of the material was found to be $G = 60$ Pa $\pm$ 3.5 Pa, and was independent of the MSG sizes used in these experiments (scale bar = 50 μm; $n = 24$). **d–e** Axial ($\varepsilon_z$) and radial ($\varepsilon_r$) strains in an MSG are swept parametrically to determine the **d** axial ($\sigma_z$) or **e** radial ($\sigma_r$) stresses associated with each combination of strain. Using these simulations, observations of MSG deformation in the radial and axial directions can be used to determine the unique combination of stresses present at that location

**Patterns of solid stress develop in multicellular spheroids**. Despite the importance of mechanics on biological function, and the widespread use of MCS cultures as model tissues for drug screening and tissue engineering, little is known about the internal force profiles generated in the MCS. Previous computational studies of internal tissue force during tumor growth predict spatial patterns of stress in a MCS-like system, in which an outer layer of cells in tension maintains the tumor mass in compression[43,49]. These predictions have recently been supported by indirect experimental evidence that tissues relax after targeted incisions[42], but the force profiles for aggregated spheroids have not been verified by direct experimental measurement. Hence, we applied our MSGs to measure force generation in MCS (~800 μm in diameter), using human bone marrow fibroblasts as a model cell line.

We used an aqueous two-phase printing technique[50] to manufacture spheroids, that places a small population of cells in close proximity to each other, allowing them to aggregate (Fig. 3a). HS-5 MCS cultures compacted over the first 24 h, after which they remained relatively constant in size (Fig. 3b). Inhibition of actomyosin contractility with blebbistatin reduced but did not eliminate MCS compaction (Fig. 3b), indicating that multiple cellular mechanisms are responsible for tissue compaction. We confirmed that HS-5 cells secrete Type I collagen by immunostaining (Fig. 3c; Supplementary Figure 13a)[51], and that any secreted ECM is not in the form of mature and relatively rigid load-bearing collagen fibers (Supplementary Figure 13b, c). Hence, incorporating soft MSGs coated with Type I collagen is unlikely to alter baseline cell or tissue contractility. Finally, we verified that over the experimental time course presented here, there were no signs of a necrotic core, either via nuclear fragmentation (see below Fig. 6a), or by analysis of carbonic anhydrase 9 (CA9), a marker for hypoxia, indicating no increased hypoxic stress (Supplementary Figure 13d, e).

MSGs were incorporated in ratios of ~1 per spheroid to minimize any impact of the sensor on MCS formation and to simplify experimental analysis. Confocal images of partially embedded MSGs confirm adhesive interactions between the

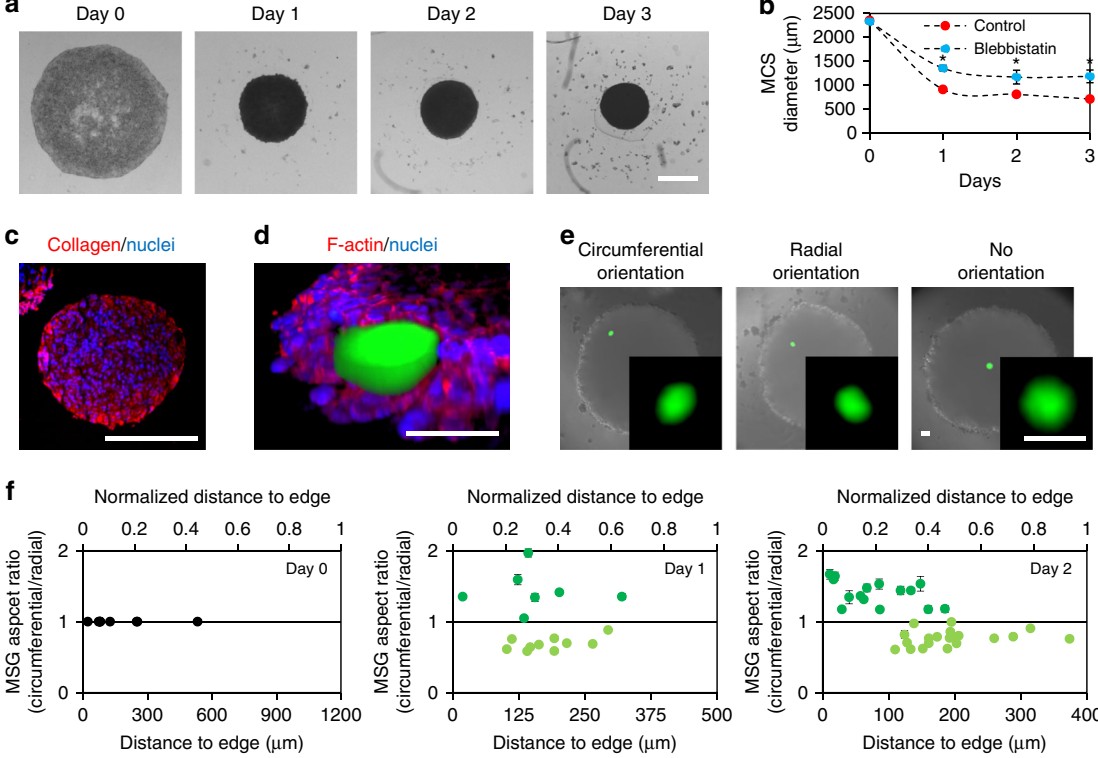

**Fig. 3** Characterization of multicellular spheroid cultures and MSG incorporation into MCS cultures using standard spheroid-forming techniques. HS-5 fibroblast cells form into a tight multicellular spheroid (MCS) over 3 days of culture. **a** Representative images of spheroid compaction (scale bar = 500 μm). **b** Quantification of spheroid size. Substantial compaction occurs over the first 24 h, and is significantly reduced when actomyosin contractility is inhibited with blebbistatin. Data reported as mean ± standard deviation, n = 9, *p < 0.05 (ANOVA with Tukey post-hoc pairwise comparisons). **c** HS-5 fibroblasts deposit collagen I over 2 days of culture in MCS (scale bar = 250 μm). **d** 3D confocal reconstruction of MSG (green) embedded at the edge of a spheroid at day 2 of culture (scale bar = 50 μm). **e** Embedded fluorescent sensors deform within the spheroid, with circumferential, radial, or no orientation (scale bar = 50 μm), based on position within the spheroid. **f** Quantification of MSG aspect ratio (circumferential MSG dimension/radial MSG dimension) reveals a spatial pattern in the orientation of the MSG deformation, with predominantly circumferential orientation (dark green) at the edge of MCS, and radial orientation (light green) towards the core by day 2 in culture. Data reported as mean ± standard deviation; n = 9, 17, and 35 for days 0, 1, and 2

MSG and the tissue (Fig. 3d). Measurements of MSG dimensions were performed immediately after seeding, and at 24 and 48 h. The spatial positions of MSGs within the spheroid were acquired using the z-coordinates (obtained from the motorized microscope stage), and the x, y-coordinates based on image position. The MCS was then disrupted with a cell lysing agent, and re-imaged to obtain the zero-stress MSG size and shape.

At day 0 in culture, and in the unstressed state following cell lysis, MSGs were the same size (Fig. 4) and consistently spherical (Fig. 3f), further confirming that mechanical properties of the MSGs are unchanged during culture. Over days 1 and 2, MSG shapes were either circumferentially, radially, or not oriented within the MCS (Fig. 3e). MSGs close to the MCS surface were imaged with high-resolution confocal microscopy (Supplementary Figure 14) and we confirmed that the expected radial and axial MSG deformation occurs in spheroids. We observed that MSG orientation correlated with position within the MCS by day 2 of culture, with circumferentially oriented MSGs being positioned predominantly towards the edge of the spheroids (as in Supplementary Figure 14) while radially oriented MSGs were located closer to the core (Fig. 3f), strongly suggesting that spatial patterns of stress evolve within the MCS structure over time. Formation of these patterns was minimized when actomyosin contractility was inhibited with blebbistatin (Supplementary Figure 15).

Next, tissue stresses were mapped based on these deformations and the zero-stress state of the MSGs (data provided in Supplementary Tables 1–4). This data reveals large compressive stresses in the kPa range in both the radial and circumferential directions within MCS cultures, with maximal compressive stress values of 1.3 kPa peaking ~150 μm into the spheroid. At the MCS edge, tensional stresses of up to 50 Pa were measured in the circumferential direction only (Fig. 4a, b). To estimate uncertainty, we considered errors in measuring the apparent shear modulus of the MSG (60 ± 3.5 Pa) and in identifying the MSG edge during image analysis (details provided in Supplementary Methods, Supplementary Figures 16, 17). Taken together, the 95% confidence interval of each measurement (presented in Fig. 4) is within ±20% of the measured value. The observed stress patterns develop as early as the first day of culture and are maintained through day 2. As expected, these average stress patterns in both the radial and circumferential directions were significantly reduced by inhibiting actomyosin contractility in MCS cultures with 50 μM blebbistatin (Fig. 4c–f).

**Stress patterns correlate to mechanical activity biomarkers**. The picture that emerges from the observed stresses is that of a circumferential tensional 'skin', only a few cell layers thick, that forms on the surface of the spheroid, and balances an inner mass under compressive loads that peak ~150 μm below the MCS

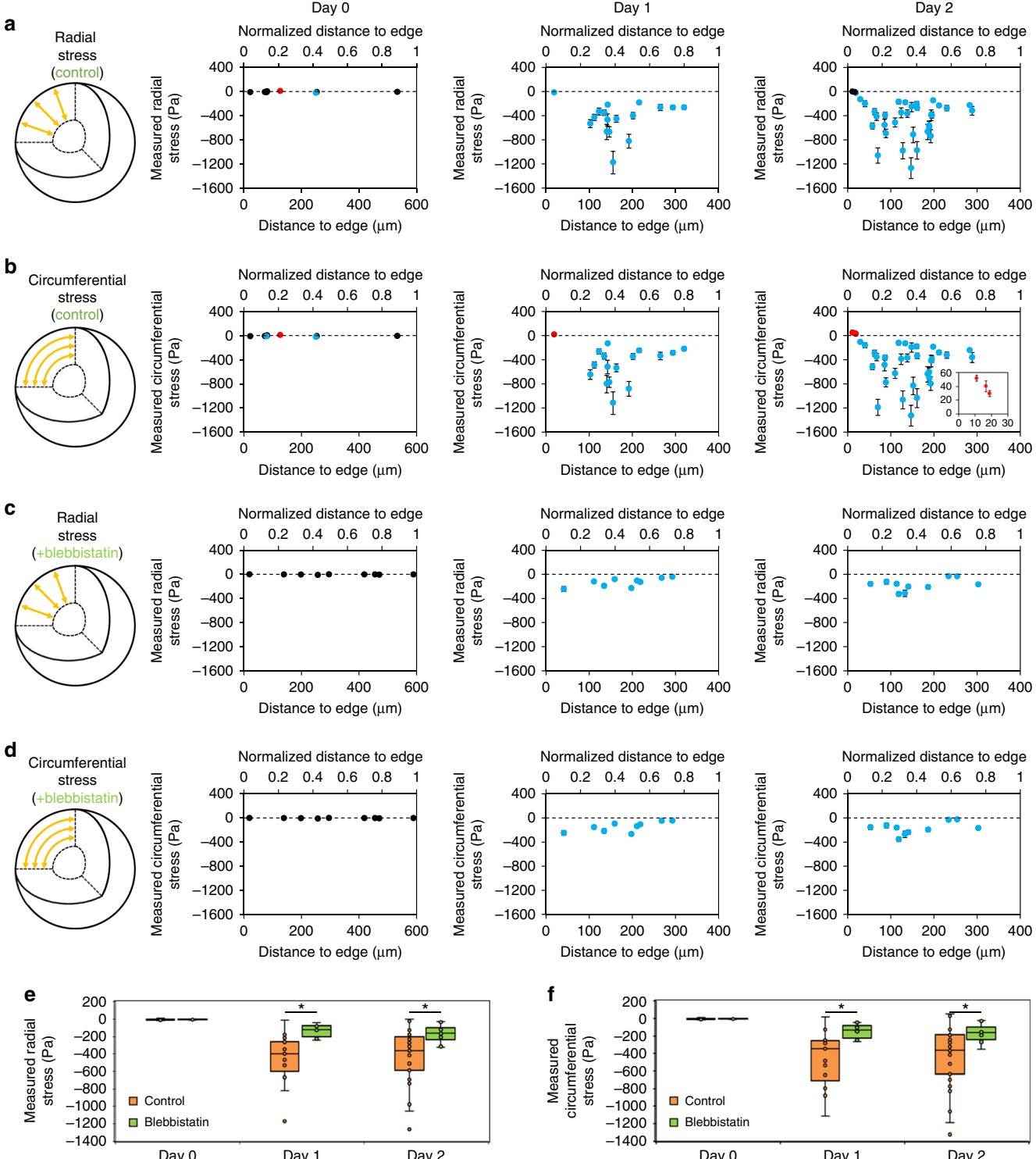

**Fig. 4** Spatial mapping of cell-generated stresses within 3D fibroblast spheroid cultures. **a–d** Internal tissue stresses in the **a** radial and **b** circumferential direction are measured at days 0 ($n = 9$), 1 ($n = 17$), and 2 ($n = 35$) of culture. By day 2, compressive stresses in the radial and circumferential directions as large as 1.3 kPa are measured within the spheroid, while (inset) tensional stresses are measured in the circumferential direction at the outer surface of the spheroid. Internal tissue stresses in the **c** radial and **d** circumferential direction are greatly reduced in MCS cultured with blebbistatin to inhibit actomyosin contractility. Red data points represent tensional stress measurements, blue data points represent compressional stress measurements, and black data points represent stress measurements close to zero (−10 Pa to +10 Pa). Error bars indicate the 95% confidence interval for each data point, determined using a combination of systemic accuracy errors in defining MSG shear modulus, and Monte Carlo simulations of stress measurement arising from precision-related errors in identifying the MSG edge. **e**, **f** The average stresses measured throughout the MCS in the **e** radial and **f** circumferential directions increased significantly from day 0 to days 1 and 2, and was significantly reduced by inhibition of actomyosin contractility. Box plots indicate the median and first to third quartile, and the whiskers span the shorter of the range or 1.5× the interquartile range. *$p < 0.05$ (one-way ANOVA with Tukey post-hoc pairwise comparisons)

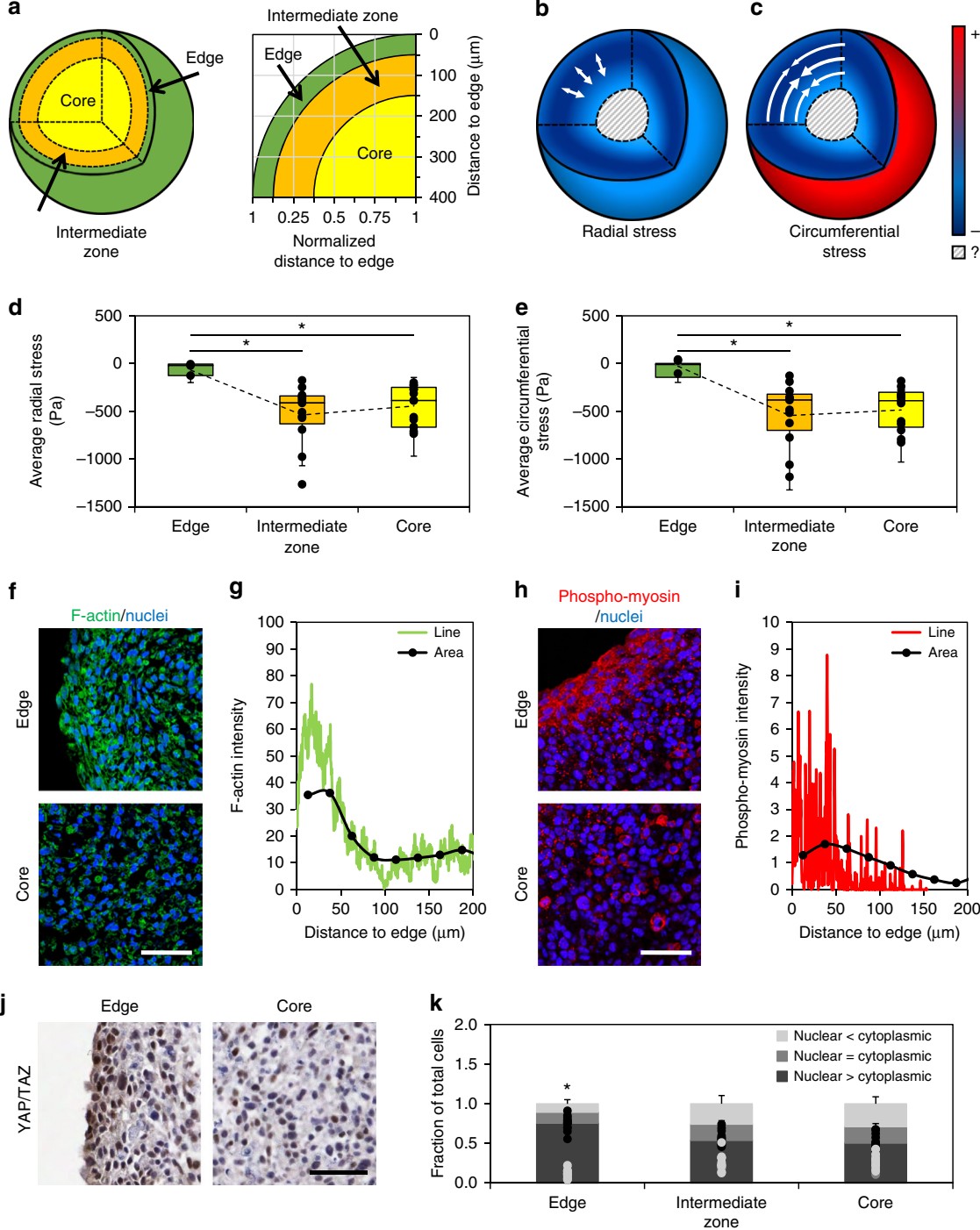

surface (Fig. 5b, c). This suggests that there may be a change in mechanical activity within the spheroid in these regions. For descriptive purposes, we considered three distinct regions of the MCS (Fig. 5a): the core region at the center of the spheroid (150–400 μm below the surface), the intermediate zone (50–150 μm below the surface), and the edge (0–50 μm below the surface). The average radial and circumferential stresses across these zones was significantly lower in the edge region where tensional stress at the surface was observed, than in the core and intermediate zones (Fig. 5d, e). A small reduction in average stress was observed in the core zone compared to the intermediate zone, but this difference was not statistically significant ($p = 0.24$ radial and $p =$

0.43 circumferential by two-tailed ANOVA), likely reflecting the large spread in cellular-scale stresses observed within the MCS.

To determine whether these patterns of observed cell-generated stress correlated with biological function, we analyzed known markers of mechanobiological activity within the spheroid. Consistent with increased tension in the edge region, cells are slightly elongated compared to cells in the core (Supplementary Figure 13f). Filamentous actin is an important component of the cellular cytoskeleton associated with tension, and was more abundant and organized at the periphery of the MCS than in the intermediate zone or core (Fig. 5f, g). Phosphorylated myosin, an indicator of active cytoskeletal contractility, was also elevated in

**Fig. 5** Characterization of mechanobiological markers within the MCS. **a** Schematic of MCS segregated into edge (50 μm from the free surface), intermediate (50–150 μm into MCS), and core zones. **b**, **c** Visual representation of **b** radial and **c** circumferential stresses within spheroid cultures, demonstrating compressional stresses in the core and intermediate zones, and a tensional 'skin' on the surface. **d**, **e** Quantification of average **d** radial and **e** circumferential stresses across the MCS at edge ($n = 5$), intermediate ($n = 15$), and core ($n = 15$) zones. Box plots indicate the median and first to third quartile, and the whiskers span the shorter of the range or 1.5× the interquartile range (*$p < 0.05$; one-way ANOVA with Tukey post-hoc pairwise comparisons). **f** Analysis of filamentous actin in sectioned spheroids reveals a highly organized cytoskeleton in cells around the MCS circumference (scale bar = 50 μm); quantified as **g** fluorescent line intensity graphs (green) and the average normalized to the selected area (black). **h** Phospho-myosin is differentially expressed at the MCS edge than in the core (scale bar = 50 μm); quantified as **i** fluorescence line intensity (red) and normalized to quantified area (black). These biomarkers suggest that spatial activation occurs in the MCS, and that the outer layer is responsible for generating contractile forces. **j** Immunohistochemistry for YAP/TAZ on sectioned spheroids shows significantly greater nuclear abundance at the edge compared to the core, indicating that the observed forces have biological significance in their ability to activate key signaling pathways (scale bar = 50 μm). **k** YAP/TAZ nuclear intensity was classified as either predominantly nuclear, predominantly cytoplasmic, or nuclear = cytoplasmic based on its relative abundance. There was a significant increase in the fraction of cells with nuclear>cytoplasmic YAP at the edge of the spheroid when compared to both the intermediate zone and the core. Data reported as mean ± standard deviation, $n = 13$, *$p < 0.05$ when compared to similar threshold levels at other spatial locations (one-way ANOVA with Tukey post-hoc pairwise comparisons)

this region (Fig. 5h, i), suggesting that edge cells are more mechanically active in generating forces. To determine whether the observed cell-generated forces are associated with known mechanotransduction pathways, we analyzed patterns of Yes-associated protein (YAP) and transcriptional coactivator with PDZ-binding motif (TAZ) activation. YAP/TAZ are transcription factors and core components of the Hippo signaling pathway, and known nuclear relays of mechanical signals from the cellular microenvironment[52]. Nuclear localization of YAP/TAZ was significantly increased only in the edge region of the MCS, under mechanical tension (Fig. 5j, k).

**Stress profiles do not arise from growth of spheroids**. The tensional skin observed in these experiments is expected for cancer tumors grown from single cells, based on observations that tumors change shape when dissected or freed from confinement[42]. The underlying premise is that the tensional skin balances an internal state of compression caused by cells proliferating within the tumor, and we asked whether proliferation plays a similar role in our aggregated MCS cultures. Since uniform proliferation in an unbounded MCS causes no additional stresses (Supplementary Figure 18), we reasoned that stress patterns can only arise if spatially distinct proliferation patterns occur. Through finite element simulations, we determined that circumferential tension at the edge and radial/circumferential compression within the MCS only occurs when cells in the intermediate zone (50–150 μm deep into the MCS) proliferate faster than cells at the edge and in the core (Supplementary Figure 18). However, analysis of proliferative state across the MCS did not support this idea. The percentage of cells expressing the Ki67 nuclear proliferation marker was greatest at the MCS edge rather than in the intermediate zone or the core (Supplementary Figure 19a, b). Cell packing densities across the MCS may also affect volumetric expansion arising from proliferation, but accounting for this factor (Fig. 6b, Supplementary Figure 19c) still did not show volumetric growth patterns that suggest proliferation patterns are responsible for the observed stresses.

**Stress profiles arise from an outer layer of contraction**. Given that the MCS cultures studied here are contractile (Fig. 3a, b), and endogenous stresses decrease by inhibiting actomyosin contractility, the observed stress profiles are likely generated by spatial patterns of compaction that create compressive stresses within the MCS. To determine whether compaction varied within the spheroid, we evaluated Haeomotoxylin and Eosin (H&E) stains of nuclei and cytoplasmic/extracellular protein content and noted that cell density is greatest on the outside of the MCS, and decreased in the intermediate zone and core (Fig. 6a, b). A higher

density of cells in the outer regions suggests that the outer shell compacts faster than the core. This is supported by increased phosphorylated myosin expression at the MCS edge (Fig. 5h, i) which indicates that the outer edge is generating cellular tension, necessary for the compaction process.

To verify that varied compaction rates in each of the zones could create the observed force profiles, we developed a finite element model of a compacting MCS with specified compaction rates in each layer (Fig. 6c–e). When compaction rates are uniform, no stress distribution exists across the MCS (Fig. 6c). When compaction occurs primarily at the edge (compressing the spheroid from the outside-in), the predicted stresses match experimental results in the edge and intermediate zones (Fig. 6d). Although not statistically significant in our experiments, we also determined that the slight decrease in stress values between the intermediate and core zones can arise if the core is mechanically more compliant than the intermediate zone (Fig. 6e), which is reasonable to expect given that the density of cells in the core region is low (Fig. 6a, b). These results together indicate that the MCS formation process itself plays an important role in setting up internal stresses, and although the stress profiles that arise are similar to those of a spheroid grown from a single cell, the underlying mechanisms that generate these stresses are distinct. Hence, the method of MCS production may play a critical, but previously under-appreciated role in defining mechanobiological activity.

**Discussion**
Measuring endogenous forces that arise in biological systems is challenging due to the material complexity inherent in most biological tissues. Extracting stresses from observed displacements requires a detailed understanding of matrix mechanical properties at the site of deformation, which may be highly non-linear, time-varying, and dependent on cell bioactivity. Even fully synthetic and mechanically defined 3D culture materials must be designed to degrade to allow cell spreading, which may locally alter mechanical properties of the tissue[53,54]. To address these uncertainties, we present microscopy-compatible probes that report quantitative, directional, and real-time measurements of stress at cellular- and supra-cellular length scales within engineered tissues. We achieve localized measurements at the length scale of individual cells (10s of μm) with a large dynamic range from 10–1000s of Pa, through the development of a compressible polyacrylamide hydrogel formulation that is soft enough to deform in response to small cell-generated stresses. Robust fabrication of microscale structures in polyacrylamide of this low stiffness required modifying the hydrogel formulation with fluorescent chain-terminating monomers to maintain the high

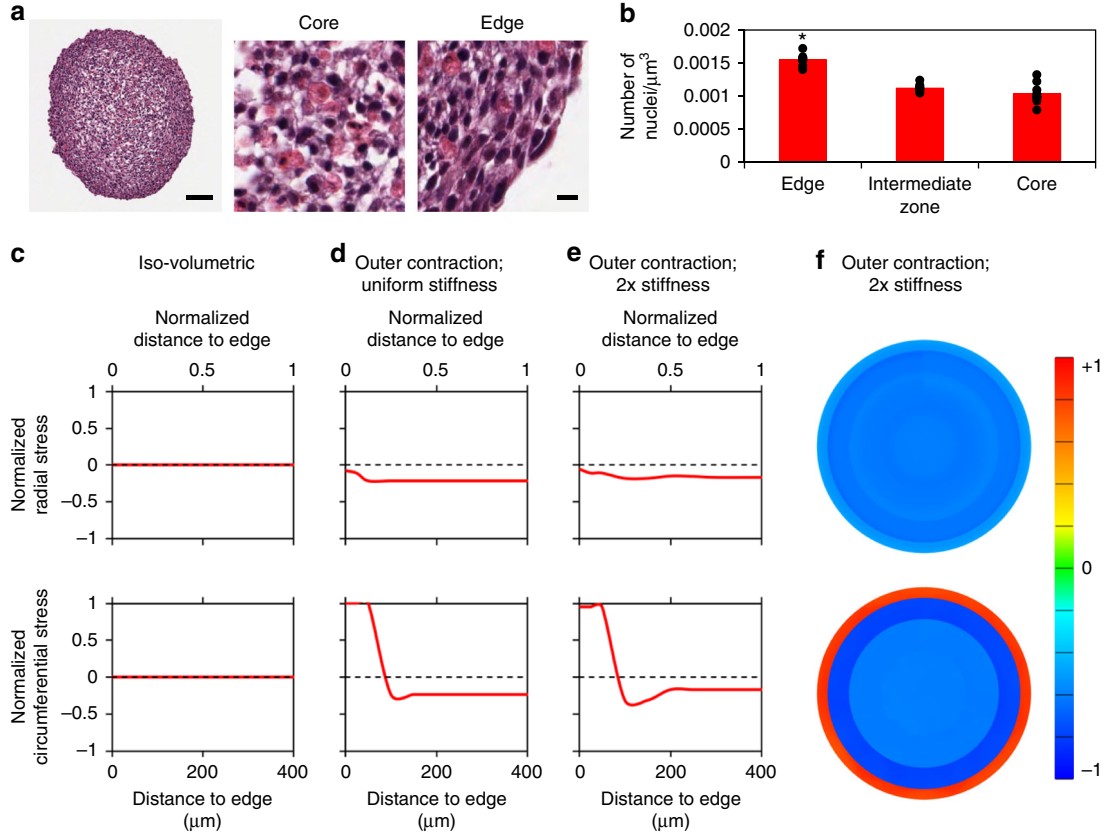

**Fig. 6** Spheroid compaction is driven by a contractile outer shell. **a** H&E staining on fixed spheroid sections indicate no cell necrosis, with greater cell density and cell elongation at the MCS edge (scale bar = 100 μm and 10 μm). **b** Quantification of nuclei packing within HS-5 fibroblast spheroid cultures indicate highest cell density in the outer layers, suggesting greater reorganization in these layers. Data reported as mean ± standard deviation, $n = 9$, *$p < 0.05$ when compared to other spatial locations (one-way ANOVA with Tukey post-hoc pairwise comparisons). **c**–**e** Finite element model developed with different rates of volumetric contraction demonstrate that simulations can align with experimental results when **e** the edge undergoes contraction with a 2x softer core compared to the edge and intermediate zones. In contrast, when only **c** iso-volumetric contraction or **d** outer contraction with uniform stiffness is present, these similar profiles are not observed. **f** Schematic of simulated stress within spheroid cultures with outer contraction and a 2x stiffness gradient between the core and the other regions

polymer-content needed for microscale polymerization, while reducing the modulus of the resulting material. Since this modification also makes the sensors fluorescent and therefore optically assayable, they can be untethered from physical or electrical connections, allowing them to be dispersed widely throughout a 3D tissue. The microspherical stress gauges (MSGs) produced could be imaged in optically dense engineered microtissues using epifluorescent and spinning disc confocal microscopes, which are readily available in many biological wet labs.

The soft MSGs fabricated allow stress measurements with as little as 10 Pa resolution, and a range of >100 Pa (tension) to several kilopascals (compression). Further reductions in polymer content to produce softer MSGs failed because MSGs self-aggregated when the surfactant was removed from the system, and the reduced fluorescent content made them difficult to find in 3D tissues. By incorporating these MSGs into multicellular spheroid (MCS) cultures, we demonstrated that directional stresses generated within the MCS range from 10s of Pascals of circumferential tension on the outside 'skin' of the spheroid, which supports kilopascals of radial and circumferential compressive stress within the inner spheroid mass. These quantitative numbers are consistent with recent studies[38,41] that indicate heterogeneous kilopascal-scale compressive stresses within cell

aggregates, but in addition to these works, provides sufficient stress-measurement resolution to capture directional stress variations across the spheroid, including the relatively low levels of tension on the tissue surface. Consistent with our force measurement profiles and proposed models for MCS formation, we found that cells located at the periphery exhibit greater contractility, as evidenced by increased expression of phosphorylated myosin, and organized filamentous actin, and that these differences in mechanical forces are associated with activation of mechanosensitive signaling pathways involving YAP/TAZ transcription factors. Speculatively, these stress profiles associated with tissue shape could hence give rise to distinct differentiation profiles and spatially-regulated biological activity.

We then determined that 'outside-in' contraction drives the MCS architecture, and gives rise to similar stress profiles as those observed in multicellular tumors grown from single cells. Computational simulations[43,49] and indirect measurements[42] have previously predicted high tensions at the edge of growing tumors, that balance internal compression caused by proliferating cell populations. We found no evidence of differential proliferation supporting this mechanism of stress generation; but instead found that contraction of an outer layer of mechanically active cells during the MCS formation process drives the observed stress

profiles. Hypothetically, high availability of oxygen and nutrients in the outer layer allows this layer to generate tension and contract, creating compression within the construct. Compressive stresses dissipate towards the less dense core of the MCS, where contractility is reduced as nutrients and oxygen are depleted. This narrative is supported by observations of filamentous actin architecture and phosphorylated myosin activation, as well as observed density and proliferative marker patterns within the tissue. Interestingly, these results indicate that similar stress patterns in aggregated MCS and those grown from single cells are generated by two distinct mechanisms, suggesting that (1) MCS formed by aggregation may be mechanobiologically comparable to tumors grown from a single cell; and (2) the tissue formation process plays an important role in regulating the mechanical state of the tissue. These findings may explain why spheroids formed using different methods could result in distinct biological functions and be difficult to compare across research labs, as described by others[55]. Hence, the spheroid formation process may be more important than previously considered, when trying to construct tissues that have mechanobiological sensitivities or functions.

Some fundamental limitations must be considered in the use of MSGs. First, because MSGs are porous hydrogel structures, they will not deform in response to altered hydrostatic fluid pressure, and can only be used to measure solid mechanical stress. Second, the remodeling capacity of the biological tissue being studied must be considered carefully prior to designing these experiments. The presented approach is only valid if the tissue remodels sufficiently to dissipate local shear stresses that arise around the MSG due to stiffness mismatches between the sensor and the tissue at the contact interface[56]. For example, if cells in a collagen matrix are stimulated to rapidly contract, the ultrasoft MSG will act as a void in the tissue and redistribute stresses around it, providing an inaccurate measurement of local stress. However, if these local shear stresses are dissipated through viscous remodeling of the tissue, deformation of sufficiently small MSGs will accurately reflect local stresses. Hence, the accuracy of this technique for a given sampling time depends on the power-law rheology of the tissue of interest, which may vary considerably from tissue to tissue[57–59]. Large multicellular spheroid aggregates are well-established to exhibit viscous deformation in response to small shear stresses within minutes[60,61], and since ECM in our spheroids is not expressed in mature load-bearing form (Supplementary Figure 13b), they are unlikely to provide additional resistance to local shear stresses over the 24 h time points studied here.

Within these constraints, we anticipate broad utility of the MSGs in furthering our understanding of the relationship between mechanical forces and biological function, as this technique can readily be extended to study a wide variety of 3D tissues. First, although we have limited our studies to coating MSGs with one ECM protein, the ability to apply the well-established sulfo-SANPAH chemistry to incorporate a wide variety of proteins and adhesive motifs allows the MSG to be tailored for specific extracellular matrix contexts without disrupting native tissue function. Similarly, mechanical rigidity of the polymer formulation achieved here is comparable to the softest body tissues[62], and can be readily stiffened by increasing the crosslinker density. Hence, MSGs can be designed to closely match the native composition and mechanics of the tissue of interest, to 'mask' the presence of the MSG and prevent the formation of a fibrotic capsule or other immune rejection response. Second, the need to obtain the zero-stress state dimensions of the MSG limits workflow. Generating monodisperse hydrogel sensor sizes using microfluidic production

strategies would eliminate the need to sacrifice the tissue for MSG analysis. Third, the current MSG technology requires analysis by optical means. This is challenging in dense tissue cultures, and we were unable to make reliable measurements more than 300 μm beneath the tissue surface, nor were we able to resolve 3D deformations at this distance. Advanced imaging techniques such as using optical windows for in vivo imaging; or excising, fixing, and clarifying tissues may be suitable. Alternatively, the MSG sensors may be decorated with brighter quantum dot-based labels, X-ray contrast agents, or acoustic contrast agents for analysis with different imaging modalities. Finally, the analysis technique developed here considers only ellipsoidal geometries for MSG deformation, and explicitly assumes that stresses within an MSG are uniform and have at least one axis of symmetry. While this is justified for analysis of supra-cellular stresses in MCS cultures, this approach does not capture sub-cellular stresses that must be present, and would not be sufficient to analyze deformation in a non-symmetric tissue. Fortunately, computational approaches to model mechanical stresses in each individual MSG can be readily developed[41].

MSGs allowed us to experimentally observe the internal supracellular stress profiles that arise within simple spheroid cultures, and to focus on how cell-generated forces contribute to heterogeneous mechanical architecture through self-driven tissue reorganization. The ability to visualize forces within biological cultures presents exciting possibilities for monitoring tissue status during tissue engineering processes, better understanding the complex interplay between cells and environmental mechanics, and developing informed strategies to coax cells into self-organizing towards appropriate tissues and organs. The overall process presented here in which MSGs were used to quantitatively map forces within tissues, followed by analytical and computational techniques to determine how those forces arise, may be leveraged to provide unique tools with which to design advanced tissue engineering strategies, in which cells are initially set up or 'primed' to mechanically construct tissues, with exquisite control over tissue structure and function at the length scale of individual cells.

## Methods

**Polyacrylamide biomaterial formulations**. Stiffness-tunable polyacrylamide (PAA) hydrogel formulations were prepared using the following acrylamide (Bio-rad, 1610140) to bisacrylamide (Bio-rad, 1610142) ratios with the stated nominal shear modulus values determined by shear rheology of bulk samples: 60 Pa (3.0 wt %/0.06 wt%); 400 Pa (3.0 wt%/0.10 wt%); 4000 Pa (7.5 wt%/0.06 wt%); 7500 Pa (7.5 wt%/0.24 wt%). To polymerize 1 mL of pre-polymer mixture, 100 μL of 1% w/v ammonium persulfate (APS; Bio-rad, 1610700) in phosphate buffered saline (PBS) and 1.5 μL of tetramethylethylenediamine (TEMED; Sigma-Aldrich, T7024) were added to initiate and catalyze the polymerization reaction. In all, 1 μL of 10% w/v fluorescein o-methacrylate (Sigma-Aldrich, 568864) in dimethyl sulfoxide (DMSO) was added to the pre-polymer mixture for the synthesis of fluorescent hydrogels.

**Microfluidic droplet generator**. A microfluidic droplet generator system consisting of a pulled circular micropipette inserted into a square glass tube[45] (Supplementary Figure 6) was used to rapidly create hydrogel microspheres of relatively uniform size. Droplets of PAA pre-polymer with the ammonium persulfate initiator and fluorescein methacrylate monomers were generated in kerosene with 6% w/v polyglycerol polyricinoleate (PGPR 4150). The TEMED catalyst was added downstream of the droplet generation site, and hydrogel microspheres were allowed to polymerize in a kerosene bath. The surfactant-rich kerosene was first replaced with surfactant-free kerosene through multiple centrifugation steps, before being displaced with PBS and allowed to swell. The process enables the rapid production of a large number of uniform hydrogel microspheres, but the large swelling ratio (Supplementary Figure 7) for soft gels magnifies even very slight variations in droplet size.

**Stirred emulsion hydrogel droplet formation**. Polyacrylamide hydrogel microspheres were fabricated using a water-in-oil stirred emulsion technique. All

solutions were prepared in clean glass test tubes (Fisher, S63288) capped with rubber septum stoppers (Fisher, FB68681). PAA pre-polymer components were syringe filtered through 0.22 µm pore size nylon filters (Fisher Scientific, 09719 C) and purged with $N_2$ for 15 min. The oil phase, kerosene (Sigma-Aldrich, 329460) with 6% w/v PGPR 4150 surfactant (Palsgaard, 90415001) was purged with $N_2$ for 30 min. PAA pre-polymer components were syringe injected into the oil phase, and the emulsion was vortex mixed at maximum speed for 10 s, generating microspheres ranging from 10 to 100 µm in diameter. The emulsion was magnetically stirred at low speed for 30 min while the PAA microspheres polymerized.

**Microsphere recovery and functionalization**. Kerosene with 6% w/v PGPR 4150 surfactant was first replaced with surfactant-free kerosene, and then recovered in PBS through multiple centrifugation steps. Recovered microspheres in PBS are no longer visible in the aqueous phase by eye as there is little polymer content. Microspheres were sterilized under UV light for 45 min, and swelled overnight at 4 °C. Microspheres were resuspended in 0.05 mg/mL Sulfo-SANPAH (G-Biosciences, BC38) in PBS and irradiated under UV light for 4 min to activate the cross-linker. Microspheres were rinsed with PBS, resuspended in 0.05 mg/mL collagen I (VWR, CACB354231) in PBS, and stored overnight at 4 °C. Collagen I coated microspheres were resuspended in PBS and stored at 4 °C until used for multicellular spheroid (MCS) culture studies. Collagen I functionalization was verified with anti-collagen I mouse primary antibody (Abcam, ab6308) and goat anti-mouse secondary antibody tagged with Alexafluor 594 (Abcam, ab150116). To account for batch to batch variability, one batch of microspheres were used for all MCS experiments.

**Shear rheometry**. Mechanical characterization of bulk polyacrylamide hydrogels was performed using a parallel plate, strain controlled shear rheometer (Anton-Paar, MCR 302). Twelve millimeter glass coverslips were placed in a shaking bath of 0.4% v/v 3-(trimethoxysilyl)propyl methacrylate (MPS; Sigma-Aldrich, M6514) in acetone for 5 min, followed by an acetone wash for 5 min. PAA pre-polymer mixture was sandwiched between 2 MPS treated coverslips and polymerized to produce hydrogel disks of 1 mm thickness. PAA hydrogel disks were placed in a shaking bath of PBS to swell overnight and to remove non-polymerized monomer. PBS volume was replaced 3 times over the course of 3 h. Excess PBS was dried off the top and bottom of the hydrogel disks and adhesively fixed between the rheometer plates. The storage modulus was measured at 10% strain from 0.001 to 10 Hz, which was verified to be within the linear elastic regime by an amplitude sweep from 1 to 80% strain at 1 Hz.

**Osmotic pressure measurement of MSG rigidity**. A long-chain dextran solution (500 kDa; www.dextran.ca) was used to extract MSG modulus from osmotic pressure-induced deformations. The system was first calibrated against a bulk disk-shaped polyacrylamide hydrogel (diameter = 13 mm) previously characterized by shear rheometry (as reported in Supplementary Figure 8). A 100 mg/mL 500 kDa dextran solution was used to exert an osmotic pressure on the bulk gel disk and the resulting radial deformation was measured. To determine the osmotic pressure created by the dextran solution, a two-dimensional axisymmetric finite element model was constructed in COMSOL v.5.3.1.201 (Comsol Inc., Burlington, MA, USA), simulating isotropic pressure-induced deformation of the cylindrical disk-shaped sample. Selection of hyperelastic material properties and meshing considerations are presented in the finite element methods section. A parametric sweep of applied pressure and the resulting strain (Supplementary Figure 10d) was used to determine that 100 mg/mL dextran solution exerts 67 Pa of isotropic pressure on the hydrogel surface. Using this pressure, a parametric sweep of MSG material properties was conducted to determine the relationship between material shear modulus and resultant deformation (Supplementary Figure 10e). MSGs exposed to 100 mg/mL dextran solution will deform to a certain extent, and this curve relates the experimental observation of deformation with the apparent shear modulus of the MSG material. Experimentally, osmotic pressure induced deformation of MSGs occurred rapidly, and microspheres equilibrated within 30 min prior to making deformation measurements. Microspheres left in dextran solution for an additional 3 h confirmed that dextran did not permeate through the polyacrylamide microspheres as no additional deformations were observed (Supplementary Figure 10c). Using this technique, the apparent modulus of the following MSGs were determined: uncoated; coated in collagen I; and coated in collagen I, cultured, and released from spheroid cultures. Measurement errors of 1 pixel in the images result in errors in estimating the apparent shear modulus of MSGs by ±7 Pa. The standard deviation in our most varied case (MSGs released from spheroids) was ± 5.3 Pa, suggesting that MSG material properties can be considered constant across each microsphere.

**Characterization of Poisson's ratio**. Polyacrylamide hydrogel strings were polymerized in 1.3–1.6 mm internal diameter glass tubes (Kimble-Chase, 34500 99) that had been pre-treated with Aquapel (www.aquapel.com) to prevent hydrogel attachment to the inside glass surface. Once polymerized, the hydrogel strings were released and hydrated in PBS for 24 h prior to mechanically stretching the strings. Hydrogel strings were stretched and observed under a fluorescent stereo

microscope (Olympus SZX16) with a LED blue/green light (Crime-Lite 2) and yellow filter. Images were captured with a trinocular-mounted Canon Rebel EOS camera. Changes in string length and corresponding changes in diameter were measured using image analysis, and used to calculate the Poisson's ratio based on the equation: $v = -\varepsilon_{diameter}/\varepsilon_{length}$.

**Finite element analysis of microsphere deformation**. Two-dimensional axisymmetric finite element models of the MSGs were developed in COMSOL v.5.3.1.201 (Comsol Inc., Burlington, MA, USA) for the purpose of quantifying axial and radial stresses associated with experimentally-observed MSG deformation. Non-linear stress-strain behavior was captured by simulating the MSG as a Neo-Hookean material with Lamé parameters calculated from experimentally determined shear modulus and Poisson's ratio values (standard simulations performed with $v = 0.3$, $G = 60$ Pa, or Lamé 1 = 60 Pa, Lamé 2 = 90 Pa). MSG compression and expansion along the axial and radial directions were simulated with strain conditions applied to the MSG solid domain. The radial and axial stresses resulting from the simulated MSG deformation were uniform through the MSG for both axial and radial stresses, and read directly from the simulations. A free quad mesh was used and optimized to ensure that the coefficient of variation was <1% (mesh element size was 2% spherical diameter; mesh element quality maintained above 0.8 for all strain combinations). A dual parametric sweep of axial and radial strains was performed to capture all possible MSG deformation combinations. The stress–strain results were then assembled in MATLAB R2017b (The MathWorks, Inc.) into two surface/contours plots using a piecewise linear interpolation fit. Multicellular spheroid MSG deformation data was then processed using the interpolation fit to obtain the associated axial and radial stresses. Additional parametric sweeps were performed for the Poisson's ratio to demonstrate the effect of compressibility on the MSG's ability to read-out stresses ($v = 0, 0.3, 0.499$, results in Supplementary Figure 12), and for variations in the apparent shear modulus of MSGs to assess for MSG sensitivity. A deviation of ±3.5 Pa in the shear modulus results in a ±5.77% error on the stress measurement (Supplementary Figure 16).

**Monte Carlo analysis of error in MSG deformation**. To obtain an accurate estimate of the error associated with MSG stress measurements, a Monte Carlo error estimation method was employed in MATLAB R2017b (The MathWorks, Inc.) using a script modified to be compatible with piecewise linear interpolation fit functions. MSG dimensions were measured repeatedly in ImageJ and the standard deviation of these measurements was used as the basis for stated errors in measurement precision. For each MSG deformation datapoint, the mean ± standard deviation in axial and radial measurements was used to generate 10,000 random data points, assuming a normal Gaussian probability distribution function for both axes. These 10,000 randomly generated strain values were then each converted to stresses using the piecewise linear interpolation fit (based on data in Fig. 2e, f), to obtain a probabilistic distribution of axial and radial stress. The resulting mean stress value and their 95% confidence intervals were used to estimate the precision error associated with measurement of deformation in each MSG datapoint (Fig. 4a–d; Supplementary Figure 16).

**Cell culture**. HS-5 fibroblasts (ATCC, CRL-11882) were cultured in Dulbecco's Modified Eagle's medium (DMEM; Fisher Scientific, 11995073) supplemented with 10% fetal bovine serum (FBS; Fisher Scientific, SH3039603) and 1% antibiotic-antimycotic (Fisher Scientific, 15240062) at 37 °C and 5% $CO_2$. The medium was changed every 2–3 days. Cells were passaged at 90% confluency.

**Multicellular spheroid cultures**. Multicellular spheroid (MCS) cultures were fabricated using an aqueous two-phase droplet printing technique by an automated liquid handler[50]. The 2 aqueous phases, poly(ethylene glycol) (PEG) and dextran were prepared as follows. 35 kDa PEG (Sigma-Aldrich, 94646) at a concentration of 6% w/v in supplemented DMEM was sterile filtered through a 0.22 µm pore size sterile filter cup and diluted with sterile reverse osmosis (RO) water at a 1:4 (PEG: water) ratio. 500 kDa dextran (www.dextran.ca) at a concentration of 20% w/v in sterile RO water was sterilized under UV light for 45 min. Solution were stored at 4 °C when not in use. To generate MCS, HS-5 fibroblasts, passaged at 90% confluency, were centrifuged at 200 g for 5 min and resuspended at a final concentration of $6 \times 10^7$ cells/mL in supplemented DMEM containing 15% v/v dextran solution. 50 µL of PEG solution was robotically dispensed into each well of a round bottom 96-well plate by an automated liquid handler (Gilson, Pipetmax). 96-well plates were pre-treated with 0.2% w/v Pluronic F-108 (Sigma-Aldrich, 542342) for 1 h to prevent cell attachment to the bottom of the plate, rinsed with RO water, and dried. 1 µL of cell suspension, containing PAA microspheres at a concentration of ~1 microsphere/µL, was robotically dispensed into each well. The plate was incubated at 37 °C in a shaking incubator at 50 rpm for 30 min to form spherical droplets. 50 µL of supplemented DMEM was added to each well, and incubated at 37 °C and 5% $CO_2$ over 2 days to form tightly compacted MCS ~800 µm in diameter. As this protocol involves the handling of micro-volumes of liquid, the use of an automated liquid handler is essential to generate MCS that are reproducible in size. To inhibit contractility, MCS cultures were treated by adding blebbistatin

(stock solutions prepared in DMSO) to the culture media immediately after seeding the cells. MCS treated with 50 μM blebbistatin (nominal concentration; Sigma-Aldrich, 203390) did not compact to the same extent as control cultures (Fig. 3b), or those treated with only the DMSO vehicle (Supplementary Figure 20). To confirm that blebbistatin did not permanently damage the contractile apparatus of the cells, it was washed out of culture, after which the MCS contracted to their baseline sizes (Supplementary Figure 20).

**Stress sensor image analysis**. MCS stress mapping imaging was performed on an inverted fluorescent Olympus microscope (Olympus, IX73) outfitted with an sCMOS Flash 4.0 Camera and Metamorph software (version 7.8.13.0). Microsphere position within MCS and microsphere shape were determined using a 10 × /0.30 NA objective on epifluorescence (Olympus, X-CITE 120 LED) on days 0 ($n = 9$), 1 ($n = 17$), and 2 ($n = 35$) of culture. MCS were then lysed in 1% sodium dodecyl sulfate at 37 °C for 1 h and the sensors imaged to obtain microsphere size under zero stress conditions. Microsphere deformation measurements were performed using ImageJ (NIH). Briefly, the multi-point tool was used to select a minimum of 3 points along the microsphere circumference and fitted with either a circle or ellipse selection. This process was repeated 3 times for each microsphere to obtain an average ± standard deviation of microsphere deformation within each MCS and following release from spheroid cultures. All images were processed at the same brightness and contrast levels. Microsphere position within each MCS was measured as the distance from the center of the microsphere to the nearest edge of the MCS in the x–y plane. This value was compared to the vertical distance from the microsphere to the bottom of the MCS using differences between the focal plane of the microsphere and the MCS. The shorter of the two distances was reported as the distance to the edge.

**Tissue sections**. MCS cultured over 2 days were rinsed with PBS and fixed with 4% w/v paraformaldehyde (Sigma-Aldrich, P6148) in PBS for 30 min. MCS were rinsed with PBS to remove remaining paraformaldehyde and stored in PBS at 4 °C. Fixed MCS were embedded in paraffin wax, sliced into 4 μm thick sections, and mounted onto clean glass microscope slides.

**Immunohistochemical analysis**. Haemotoxylin and eosin (H&E) staining was conducted following standard protocols. Cell elongation was measured by quantifying cell length manually from these stained tissue sections using ImageJ. A random selection of cells from the edge and core regions were measured over 3 spheroid sections ($n = 30$). Antigen retrieval of deparaffinized spheroid sections was performed in boiling 10 mM citrate buffer at pH 6.0 for 10 min. Spheroid sections were blocked for 5 min with Universal Blocking Reagent (BioGenex) at room temperature. Primary anti-Yap/Taz antibody (Cell Signaling, rabbit monoclonal, 1:100) was diluted in blocking solution, anti-Ki67 (Ventana, rabbit monoclonal), and anti-CA9 (Ventana, rabbit polyclonal) were used as per manufacturer's instructions. Slides were incubated with primary antibodies overnight at 4 °C. Immunohistochemical labeling was detected using SignalStain® Boost IHC Detection Reagent (Cell Signaling) and developed with SignalStain® DAB Substrate Kit (Cell Signaling). Spheroid sections were counterstained with hematoxylin. Stained sections were imaged using Aperio ScanScope XT slide scanner, using 20 × objective, and analyzed using Aperio Spectrum. Resulting stained sections were reviewed by an experienced comparative pathologist (M.C.G.). IHC Nuclear Staining algorithm was used to quantify the fraction of cells exhibiting varying Yap/Taz nuclear staining intensities and the fraction of cells positive for Ki67 nuclear staining across 3 spheroid layers of the following widths: edge 25 μm, intermediate zone 50 μm, and the remaining inner core. Positive Pixel Count algorithm was used to quantify the average intensity of all pixels of CA9 staining in 25 μm outer layer and the remaining core. Spatial positions were scaled based on the observed shrinkage of the tissues during the fixation processes (factor of ~2), and reported as dimensions of the original tissue.

**Immunofluorescence analysis**. Immunofluorescence staining of phospo-myosin was performed following a previously described protocol[63]. Paraffin sections were deparaffinized and boiled in Tris-EDTA buffer (20 mM Tris, 1 mM EDTA, 0.05% Tween-20 at pH 9.0) for 7 min in a pressure cooker. Sections were blocked in 10% normal goat serum for 1 h followed by overnight incubation of primary antibody at 4 °C. Sections were incubated for 1 h with the appropriate secondary antibody conjugated to Alexa546 (1:750 dilution). The primary antibody used was an anti-phospho-myosin light chain 2 (pMLC) (1:200, CST #3671). Actin filaments were stained with 0.1 μM phalloidin for 15 min. Confocal imaging was performed on an LSM700 from Zeiss with 20 × /0.8 NA or 40 × /1.4 NA lenses.

**Second harmonic generation imaging**. SHG images were acquired on a LSM710 Multiphoton (Carl Zeiss Canada) equipped with a Plan-Apochromat 20 × /0.8 NA, 800 nm direct-coupled multiphoton laser with or without dispersion compensation – Chameleon (Coherent), and SHG signals from Ch1: 493–630 were recorded with PMT detectors with gain of 750 V.

**Volumetric growth calculations**. Volumetric growth calculations were performed based on Ki67 sections. All images were analyzed in 3 distinct regions: edge, intermediate zone, and core. Ki67 images were analyzed for white content to determine the area not occupied by cells. These values were subtracted from the area of each region and divided by the number of nuclei to obtain the area occupied by a single cell, which was then converted to the volume occupied by a single cell assuming spherical geometry. Ki67 results were used to determine the number of dividing nuclei. This value was multiplied by the number of initial nuclei to determine the number of new nuclei following a cycle of cell division. Volumetric expansion was defined as the ratio of the new volume (new total nuclei x volume of a single cell) to the old volume.

**Finite element modeling of tissue sphere formation**. Finite element models of the tissue MCS were constructed in FEBio as a solid sphere containing three layers with distinct mechanical properties[64]. The layer thickness from the surface to the core of the sphere was 50 μm (edge), 100 μm (intermediate zone), and 350 μm (core), respectively. The active contraction of the cell population was represented by a solid mixture material consisting of a neo-Hookean solid component and an isotropic contractile stress component[65]. The compressive neo-Hookean solid allows compaction of the tissue microsphere model under active contraction. Three contraction and stiffness conditions were modeled for the tissue microsphere and the circumferential and radial stresses for these conditions were compared. In the first case, all three layers of the sphere model were assumed to have the same stiffness and undergo the same amount of active contraction (isovolumetric). In the second case, all three layers of the sphere model were assumed to have the same stiffness with only the outermost layer undergoing active contraction (outer contraction; uniform stiffness). In the third case, the stiffness of the outermost layer and the intermediate layer of the sphere model were assumed to be 2 times higher than that of the core layer, with only the outermost layer undergoing active contraction. Tissue MCS geometry was discretized by about 8800 3D quadratic tetrahedral elements capable of large deformation.

**Statistical analysis**. Comparative data analysis of populations was performed without pre-specifying a required effect size. Datasets that were normally distributed, with similar variances between compared groups were analyzed using one-way ANOVA; or ANOVA based on ranks to test for significance in the three layers of the MCS (edge, intermediate zone, and core), and for characterization of MSG apparent shear modulus. Post-hoc pairwise comparisons were conducted using the Tukey's method. All statistical analyses were performed using SigmaStat 3.5 (Systat Software Inc., San Jose, CA, USA).

## Data availability
Data supporting the findings of this manuscript are available from the corresponding author upon reasonable request.

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

## Acknowledgements

We thank Marie-Christine Guiot, Dongmei Zuo, and Camilla Jiang for expertize and technical assistance; CMC Microsystems for access to simulation software; and David Juncker, Allen Ehrlicher, and Edmond Young for critical reading of this manuscript. This work was supported by the Canadian Cancer Society (Grant # 704422), NSERC Discovery RGPIN-2015-05512 (C.M.), NSERC Discovery RGPIN 261938-13 (R.L.L.), NIH R01-HL085339 (A.J.P.), and NIH CA196018 (S.T.). We gratefully acknowledge support from the NSERC Post-graduate doctoral fellowship to W.L., and the Canada Research Chairs in Advanced Cellular Microenvironments to C.M.

## Author contributions

C.M., S.T., and A.J.P. developed the original conceptual approach for this work. W.L., R.L.L., and C.M. designed the experiments. W.L. performed MSG sensor measurement experiments and analysis. W.L. performed mechanical characterization. N.K. performed MSG deformation finite element simulations. S.M., R.H., and E.K. conducted biomarker staining, imaging, and analysis. S.T., A.J.P., L.M., M.P., R.L.L., and C.M. contributed equipment, reagents, and analysis expertize. R.Z. developed the spheroid formation

computational simulations. W.L., R.L.L., and C.M. drafted the manuscript. All authors edited the manuscript.

## Additional information

**Competing interests:** The authors declare no competing interests.

