## [Peer Review File · Nature Communications]

Reviewers' Comments:

Reviewer #1:

Remarks to the Author:

Lee and colleagues have developed soft hydrogels spheres to record mechanical stresses, including isotropic ones, applied by cells within aggregates. They identify anisotropic stresses at the surface of aggregates, which they propose to be caused by cell contractility rather than cell proliferation. The new tool developed by Lee and colleagues provides an alternative to those already available. This alternative is promising as it allows the measurement of stresses at magnitudes that current methods cannot reach. The current manuscript provides only one example of what this new method can achieve and doesn't describe or explore enough the potential and limitations of this new method.

1-In Figure 3d and e, there is only one measurement of the stress in the core of the aggregate. From this single measurement, the authors conclude that the stress plateaus. This is clearly not sufficient. In addition, the authors fit a dashed line on the data. I could not find the description of this fit. The fit on Figure 3d would look extremely different without the single measurement at the core. The fit on Figure 3e doesn't seem fairly justified.

2-The authors argue that surface contractility is responsible for the surface radial and circumferential stress. This is based on acto-myosin localisation. Can the authors inhibit acto-myosin contractility using for example, cytochalasin and blebbistatin? This would provide stronger evidences in favour of their explanation.

In addition, cells seem to orient along a circumferential axis at the surface of the aggregate. That would be in line with the authors hypothesis. Can the authors measure cell circumferential alignment along the radial axis of the aggregate?

The authors exclude cell proliferation as a source of tissue stress. This is based on the distribution of proliferation pattern. Can the authors interfere with cell division (using for example inhibitors of replication) to actually test this hypothesis?

3-In Figure 3, the authors show the normalized diameter of the aggregates over time. The normalization prevents the reader from knowing the actual size, and its variation, of the aggregates. Can the authors show the actual size with an estimation of the variability (mean \pm standard deviation)? This is very important as the authors go on to categorize regions of the aggregates as "the core region at the center of the spheroid, the intermediate zone (50 to 150 μ m below the surface), and the edge (0 to 50 μ m below the surface)". It is difficult to evaluate this categorization without the actual size measurement.

4-The authors describe briefly how the addition of the collagen layer on top of the MSG could affect the measurement. Actual measurement of the thickness and uniformity would be good to have. Also, they estimate the elastic modulus of the collagen layer but I could not find how this is calculated. Finally, to actually know how this affects the measurement, varying the thickness of the collagen would be a good test of their system.

Also, on page 6, authors claim that "mechanical properties of the MSGs can be considered constant even after embedding in the tissue of interest". However, the controls on collagen distribution appear to have been done on MSGs outside of tissues (Figure 1). If that's right, the claim should be changed.

5-The authors explain how reducing the stiffness of MSGs increases the resolution of the measurement. Can they discuss how low in stiffness and resolution their system can go? What happens to the MSGs when they are too soft?

The authors describe the limitations of previously developed methods to measure the stress applied by tissues. More details on the limitations of this new technique would be fair. What is the range of stress that can be measured? What types of tissues can it be used with? Can other ECM components be used? The authors bring forward the advantage of being able to measure isotropic

stress but make little use of this measurement. What context would that be useful for?

6-In Figure 3c, the authors show 3 inserts with MSGs at different locations of the aggregates. This gives the impression that multiple MSGs are placed within one aggregate, which is not the actual experiment, as far as I understood. This is giving the reader the wrong idea. Can the authors change to show 3 actual aggregates with the MSGs as an overlaid image? This will show the MSGs in their actual context. Moreover, a high-resolution image of the cells around the MSG would be very good to evaluate how cells interact with the MSGs.

7-Images of the aggregates in Figures 4 and 5 show many gaps in the tissue, in particular at the core. Is that correct? How much fluid is there? Is this fluid under high pressure? Is that fluid also exerting pressure on the MSGs?

8-On page 7, "HS-5 MCS cultures compacted dramatically over the first 24 hours". Maybe "dramatically" is exaggerated.

Reviewer #2:

Remarks to the Author:

In this manuscript, Lee and colleagues take on the important challenge of measuring cell-scale multi-directional forces within living, 3D tissues. Previous approaches to measuring these stresses have their limitations, as well described. The authors use soft polyacrylamide hydrogel particles, termed as microspherical stress gauges (MSGs), embedded in multicellular spheroid cultures to measure stress by monitoring changes in shape and volume of the particles. The particles are compressible and therefore can report on hydrostatic stresses, giving this approach a distinct advantage over a previously established oil droplet deformation based technique. There are some general similarities with a manuscript published by Dolega and colleagues in Nature Communications, in which deformation polyacrylamide particles report on stress in a spheroid under external compression. However, there are some distinct differences: in this manuscript softer particles are used, which are therefore more sensitive to stress, and no external loads are applied. Using this approach, the authors report an interesting and unexpected profile of radial/circumferential tension/compression within contracting spheroids. Cells at the periphery of the spheroid are enriched for YAP/TAZ exhibit signs of actomyosin contractility. Given that the Dolega paper was published in Nature Communications very recently, I think this manuscript is suitable in scope and impact for Nature Communications. However, there are a number of issues that need to be addressed before this manuscript is suitable for publication

1. The polyacramide gel particles exhibit an elastic modulus of 100 Pa, which is far below the typical range used for PA gels. This would suggest that the particles might be highly viscous. While the loss modulus is reported at 10 Hz, a frequency sweep for the storage and loss modulus should be reported. If the loss modulus: storage modulus ratio is much higher at low frequencies (0.001 Hz), that could impact interpretation. This is important to understanding the capabilities of these stress sensors.

2. The mechanical properties reported appear to be based on bulk rheological measurements. This should be complemented by AFM measurements of the bead elastic modulus to confirm that beads (or another micro-scale technique), formed through a distinct process relative to the bulk gel, exhibit similar properties to the bulk gel.

3. The MSG sensors are coated with collagen. This is an unusual choice, as cells actively sense and respond to collagen-coated hydrogels (i.e. exert traction forces on the gels in a stiffness dependent manner). As such, the beads may induce cells to actively deform them and thus the measurements are not passive readouts of stress in the spheroid. The authors should also conduct and report measurements with uncoated gels, Given that the particles are embedded in the multicellular spheroids, this should provide a passive readout of local stresses. In addition, or alternatively, the authors could probe stresses measured when the gauges are coated with a cell-

cell adhesion ligand (i.e. the n-cadherin mimetic ligand like HAVDI – see Cosgrove, et al., Nature Materials 2016) as a better mimic of the stresses that a cell would experience in a spheroid.

4. The data in Figure 3d and 3e, upon which the conclusions about the stress profile are based, are not very convincing. There is a large scatter in the data, and the data are fit with some hand-drawn dashed line. A few suggestions here: first, the authors should normalize the x-axis to be the distance between the edge (0) and the center of the spheroid (1). This makes the results easier to interpret as the beads are likely of different size. Second, the authors likely need a much higher number of measurements here to have any confidence in the stress profile, given the scatter (for example in 3e at 0, one of the triangles is measured at 0 and the other is at 115 Pa; and there appears to be only one data point near the center of the spheroid). Third, the authors need to figure out the appropriate way to support their conclusions with statistical tests – perhaps a comparison with the day 0 profile could be instructive here? Finally, the authors should include the raw data for bead extension and orientation vs position in the sphere that 3d and 3e are based on.

5. The process studied here – of multicellular spheroids formed and then allowed to contract over 3 days is distinct from the often-studied case of spheroids allowed to proliferate from a single cell starting point (i.e. Helmlinger, et al., Nature Biotechnology, 1998). What the authors are studying is a contractile process as cells contract as they form cell-cell junctions, and not a spheroid expansion process due to proliferation (there is probably not much cell division over 3 days). While there is no problem with this approach, the authors should be careful about making broad conclusions. For example, there is no reason that it would be expected that “compression [would be] induced by proliferative growth” as written in the abstract as the spheroid is still in the contractile phase. Similarly, the statements, “did not support the conclusion that tension at the edge is caused by proliferative expansion” and “Our results confirm, for the first time by direct experimental measurement, computational simulations that predict high tensions at the edge of tissues on patterned substrates of varying geometries⁵¹ and within growing tumours^{42,48}, as well as other indirect measurements of internal stress such as the degree of tissue relaxation following release of internal solid stresses⁴¹” should be qualified, as these contracting spheroids (which decrease in size) are likely to be different than growing tumors.

6. The YAP/Taz result is potentially interesting, however, the stains are somewhat difficult to see and interpret. The standard in the field is to use immunofluorescence and to quantify the ratio of nuclear: cytoskeletal YAP. The authors should do this.

7. The comment “Consistent with several studies^{51,52}, these results suggest that the spatial patterns of endogenous forces that arise within 3D clusters of aggregated cells could play an important role in specifying stem cell lineage commitment” is highly speculative, and should be moved to discussion section.

Reviewer #3:

Remarks to the Author:

General Comments:

The authors developed a new technique for measuring local stresses in 3-D multicellular tissue constructs. The technique is based on embedding small, solid hydrogel spheres of low stiffness into the tissue. Based on the measured deformation of the spheres caused by cell generated-stresses and sphere stiffness, the authors calculated stresses in the vicinity of the sphere. The advantage of the proposed technique over previous similar techniques (e.g., oil droplets) is that the hydrogel spheres are elastic and compressible solids and thus sensitive to both shear and volumetric stresses. While this is a very interesting strategy for measuring local stress in a 3-D living environment, there is a major concern about how accurate estimates of the local stress are for the following reasons.

It is well known that the stress field surrounding inclusion in an elastic matrix depend on many factors, including the mismatch of the elastic moduli of the matrix and inclusion, the location of the inclusion, the boundary conditions at the inclusion-matrix interface (perfect bonding or sliding) (cf. Jasluk et al. J Appl Mech 1997). Since the stiffness of the embedded spheres does not match the stiffness of the surrounding multicellular spheroid the estimates of stress in the spheroid may not

be accurate. This mismatch is further accentuated by the finding that there is a stiffness gradient in the radial direction of the spheroid. Furthermore, it is not clear what are boundary conditions at the sphere-spheroid interface; is it a firm bonding or there is sliding, which also bears effect on the stress field surrounding the sphere. If I understood correctly, the authors argued that the above concerns are valid only for inert matter: "For static structures in which MS is rigidly coupled to the surrounding matrix, and the matrix does not have the opportunity or capacity to reform and remodel, MSG readouts will be coupled to the stiffness of the surrounding tissue." (l. 287-289). However, since the spheroid is a living matter "in which cells are free to undergo plastic reorganization through morphological adaptation and migration during spheroid formation" (l. 286-296) stiffness mismatch between the spheres is not an issue. I am not convinced that this is true. Some stronger argument/evidence is needed.

Minor Comments:

l. 116-120, Fig. 2a: To the first approximation, the shear modulus of the hydrogel appears to be constant over a broad range of strains (i.e., linear elasticity). However, a more detailed inspection of the data in Fig. 2a reveals that at low strains the modulus exhibits a short-range softening, followed by a long-range stiffening, indicative of non-linear stress-strain behavior, which is expected for compressible materials. If the gel were incompressible (e.g., rubber-like), then constant shear modulus over a broad range of strains is a likely behavior.

What is the Poisson's ratio of the hydrogel?

l. 178-179, Fig. 3e: "At the MSC edge, tensions of up to +30 Pa were measured in the circumferential direction only." The data in Fig. 3e show tensile stresses at the edge that are > 100 Pa.

l. 238-241, Fig. 5e: What is the stiffness gradient in the spheroid that is required for the stress distribution obtained from the FE model to be consistent with the experimental data? How is the stiffness gradient related to the stress gradient? It has been shown previously that cells and tissues of different types have the shear modulus directly proportional to the state of endogenous stress (cf. Canovic et al. Biomech Model Mechanobiol, 2014). Do the stress and stiffness distributions in the spheroid follow a similar trend?

Reviewer #4:

Remarks to the Author:

In the work by Lee et al, ultra-soft hydrogel beads were fabricated and used as a microspherical stress gauge (MSG) for evaluation of the stresses generated during cell spheroid aggregation. The macroscopic/coarse-grained distribution of the stress field within a spheroid were calculated using experimental measurements in conjunction with computer finite element simulations, which together suggested formation of a tensional skin around the spheroid. The manuscript is well structured and clearly written. However, from a solid mechanics perspective, this reviewer has some major concerns that preclude recommending publication of the manuscript in its current form.

Major:

1. The main novelty of the work is claimed to be replacing incompressible oil droplets (ref. #38) with compressible hydrogel particles that exhibit an isotropic linear elastic mechanical behaviour. While Poisson's ratio (ν) is the major factor controlling volumetric changes in such compressible materials; the authors fail to account for this quantity in the manuscript.
2. In relation to this, in section "MSG mechanical characterization ..." on page 5, the shear modulus of $G=55$ Pa is mentioned while on pages 3 and 5 elastic modulus of $E \approx 100$ Pa is stated. Considering $G=E/[2(1+\nu)]$, where ν is the Poisson's ratio, implies $\nu \approx -0.1$ implying MSGs would dilate as they experience compressive stresses. Nevertheless, in Fig. S10, the results of three simulations with $\nu=0, 0.3$, and 0.499 were presented and it is mentioned in the caption that incompressible materials yield a singular matrix. At the end, it is unclear which value was adopted for Poisson's ratio ν (any of $0, 0.3$ or a negative value might have been considered).
3. On page 3, the authors state: "their [oil droplets'] incompressibility does not allow isotropic

deformation". However, it is unclear why only shear rheometry was employed to extract the shear modulus. Since calculation of Poisson's ratio is essential in this study; dilatational measurements in addition to shear rheometry can yield an estimation of both shear and volumetric mechanical properties and would be highly recommended.

4. Authors considered linear superposition to derive components of the stress tensor from measured components of the strain tensor. This principle is only valid for linear elastic materials. However, as depicted in Fig. 2 and Fig. S10 the stress-strain relationship is non-linear. This observed behaviour (for linear elastic materials) is normally referred to as geometrical nonlinearity (not material nonlinearity) which arise from the spherical shape and occurrence of large strains within the linear elastic MSGs. Therefore, the overall behaviour of MSGs can be non-linear, and application of superposition principle would give errors in estimation of stress components. These issues need to be discussed in a rational and complete manner.

5. In the abstract, "The Pascal-scale sensitivity of this technology can capture very small stresses ..." is mentioned. However, based on data shown large errors can exist in the presented calculations. For instance:

a. As shown in Fig. 2a, material parameters are not constant numbers due to errors in estimating these parameters.

b. From Fig. 3c, it is unclear how accurately the periphery of the MSGs could be determined. Consequently, there would be an error in measuring its dimensions, and consequently in estimating strains.

c. Application of the linear superposition principle for non-linear materials would also give rise to errors. Furthermore, due to the non-linear geometry, the stress-strain curve is not symmetric with respect to the origin. Therefore, in this case there is a stress-induced anisotropy which was not clearly acknowledged.

d. Considering above sources of error, it is recommended that the authors perform error analysis and report the actual sensitivity of their mechano-sensors.

6. Referring to Fig. 2, the finite element simulation is not well explained in the manuscript. For example, there is no information about the element type, mesh size, and whether effects of non-linearity (definition of large strains or large deformation) have been taken into account.

7. Figures 3d and 3e show that stresses as high as ± 100 Pa have been calculated. However, the magnitude of stress in the stress-strain curves represented in Figures 2c and 2d (or Figure S10) was not extended to such high values. Therefore, estimations of ± 100 Pa are questionable and in the results of FEM simulations the stress axis needs to be extended so it contains the ± 100 Pa range.

8. In relation to this comment and with regard to Fig. 3d and Fig. 3e, for a MSG located near the edges of the spheroid (distance to edge is a few micrometres) the radial and circumferential stresses take values near -100 Pa and $+100$ Pa, respectively. In other words, with that stress and FEM data, we expect that the MSG changes from a spherical to ellipsoidal shape due to a compressive stress of 100 Pa acting along the minor axis of the ellipsoid and a tensile stress of similar magnitude acting along its major axis. Conservatively, we ignore the circumferential stress and assume that only radial compressive stress is being applied to the ellipsoid. By extrapolating stress-strain curves in Fig. 2d or Fig. S10a or S10b (the $\Delta r/r_0$ - σ_r curves), at $\sigma_r=100$ Pa, the strain will be more than ~ 1 , which is not possible. In other words, accepting Fig. 2 or Fig. S10 as a valid constitutive law is inconsistent with observed/calculated stresses as high as 100 Pa. Furthermore, Fig. 3c shows a slightly deformed ellipsoid (egg shape) near the spheroid edge, which seems inconsistent with the stress-strain relationship presented; it should be much more elongated in that region.

9. Why are the results presented in Figure S11 (showing non-linear stress-strain curves) different from those in Fig. S10 (showing linear stress-strain)? Furthermore, what is the difference between Fig. S11a and S11b?

10. Fig. 3d shows 12 data points while Fig. 3e 17. The authors should mention clearly how many MSGs and how many spheroids were used to draw Figs. 3d and 3e.

11. Authors should use the results of more tests to fill the gap between $250 \mu\text{m}$ and $500 \mu\text{m}$ in Figs. 3d and 3e. Moreover, referring to Fig. 3e at close distances to the edge (distance ≈ 0) only one data point exists (with $\sigma_\theta \approx 110$ Pa). More data are needed close to the edge.

12. The authors used an "Olympus, IX73" microscope which is presumably an epifluorescence microscope. However, it would be much more informative if the authors would use a confocal microscope and present 3D reconstructed images of an ellipsoidal MSG and a stress-free spherical MSG obtained after lysing the spheroid. Such images could be added to Fig. 3. Furthermore, the authors indirectly assumed, probably due to symmetry of the spheroid, that the two circumferential components of stress are identical. This assumption needs to be verified and it is essential to show a 3D image of the ellipsoid with two axes of similar length.

13. Referring to Fig. 5b, the higher stiffness of the spheroid at its edges is justified and correlates with higher cell density at the outer layer. However, in order to quantify the stiffness of different regions within the spheroid, it would be much more reliable to use a direct method such as atomic force microscopy and report the real stiffness magnitudes.

14. Forces acting inside the spheroid have a dynamic nature and their magnitude and direction change as a function of time and during spheroid formation. The authors measurements were only carried out at a particular time on the second day of culture. However, it would be interesting to measure the variation of radial and circumferential stresses at a specific location versus time (for instance, at day 1, 2 and 3). This would enable the authors to fairly assess the power of the proposed method to assess the "stresses arising during multicellular spheroid aggregation" as mentioned in the abstract.

15. The authors neglected the possible effects of time on the mechanical behaviour of the MSGs. The employed PA gels are known to be elastic, however for the low modulus material used in this study, the authors should provide evidence for the assumed constant elastic mechanical behaviour (non-plastic deformations) over the 3 days of experiment. This could be simply done through lysing the spheroid and showing the images of MSGs (preferably those located at the edges where stress/ strain is high) at day 3.

Minor:

- On page 6, the authors mentioned: "However, when the tissue can plastically reorganize and flow around the MSG, as in the MCS culture models presented here, stresses will be redistributed to minimize mismatch-driven variations". Authors should provide more evidence to support this statement.
- On page 9, line 13, a sentence started with a lower-case letter (finite element models...).
- On page 13, last line, "Dolega et al" the point is missing (Dolega et al.).

We thank the reviewers for their time, effort and insight in examining our manuscript. We agree with the issues and concerns raised, and have conducted several additional experiments, developed new techniques and analyses, and restructured our critical discussion of this work. The result, we feel, is a substantially more rigorous and insightful manuscript.

Overall, we have made the following core improvements to the experiments and analysis:

- 1) Development of a new finite element model to determine stresses from MSG deformation that does not rely on linear superposition, and accounts for viscous flow in the cell spheroid.
- 2) Mechanical characterization of shear modulus and poisson ratio of the bulk polyacrylamide hydrogel formulation, as well as the apparent shear modulus of individual mechanosensors.
- 3) Confirmation of the validity of several core assumptions pointed out by the reviewers, including a detailed analysis of sources of error in the system
- 4) Extension of these studies to monitor stress evolution over time, and in response to inhibitors; as well as collecting more data to support our conclusions.

Our new findings and responses to the critiques are outlined in the following point-by-point response.

We would like to thank the reviewers and editors for what has been an extremely rewarding peer review process. The work is a substantial improvement over our previous submission, and we hope that this revision satisfactorily addresses all the reviewers' concerns.

Reviewer #1 (Remarks to the Author):

1a-In Figure 3d and e, there is only one measurement of the stress in the core of the aggregate. From this single measurement, the authors conclude that the stress plateaus. This is clearly not sufficient.

[Note: similar response to Reviewer 2, point 4b; and Reviewer 4, point 11.]

We agree that the single point is not sufficient. To add more data, we attempted additional experiments, and although we improved our sample size (Fig. 3f; Fig. 4), we could not image MSGs deeper than 300 μm below the MCS surface, likely due to the optical density of the spheroids. We attempted two-photon confocal microscopy but were unable to penetrate further without unacceptable signal losses. Hence, we removed the single data point as it could not be repeated at this depth. Our additional data indicates that there is a small decrease in the average stress in the core region compared to the intermediate zone, but this difference is not statistically significant ($p = 0.24$ by two-tailed ANOVA), likely due to the high spread in measurement values. Hence, this does not allow us to conclude that the stress plateaus. We have therefore restated this observation, incorporated our new data and analysis, and explicitly acknowledge the limitation of this fluorescent optical imaging modality in the discussion section of the manuscript. The relevant updated datasets and analysis are reproduced in this response.

Selected panels from Figure 4 and Figure 5 (updated). Spatial mapping of cell-generated stresses within 3D fibroblast spheroid cultures, demonstrating additional data points collected within the spheroid. Internal tissue stresses in the (a) radial and (b) circumferential direction are measured at days 2 of culture (Figure 4). Using the updated analysis models developed in response to reviewer comments, compressive stresses in the radial and

circumferential directions as large as 1.3 kPa are measured within the spheroid, while (inset) tensional stresses are measured in the circumferential direction at the outer surface of the spheroid. In Figure 5 (d, e) stress measurements are grouped based on location for statistical comparison. * indicates $p < 0.05$ (one-way ANOVA with Tukey post-hoc comparisons).

1b-In addition, the authors fit a dashed line on the data. I could not find the description of this fit. The fit on Figure 3d would look extremely different without the single measurement at the core. The fit on Figure 3e doesn't seem fairly justified.

We apologize for not clearly describing the line. The dashed line was not intended to 'fit' the data, but was used to simply draw attention to the envelope shape of the curve, as was done by Campas et al. in their oil-droplet publication. We recognize that this approach is unconventional, and have removed it. We considered developing a "fit" based on an n-order polynomial expression, but the parameters obtained from such a fit would have little physiological or physical relevance. Hence, we have elected to remove these lines from the image to avoid confusion, and have instead used average stress results within the specified zones to convey patterns of mechanical stress within the spheroid. This new analysis has been included as Fig. 5d, e (see figure in the response to previous question), and the results and discussion section updated accordingly.

2a-The authors argue that surface contractility is responsible for the surface radial and circumferential stress. This is based on acto-myosin localisation. Can the authors inhibit acto-myosin contractility using for example, cytochalasin and blebbistatin? This would provide stronger evidences in favour of their explanation.

To address this comment, we conducted spheroid formation experiments while inhibiting acto-myosin contractility with 50 μM blebbistatin. As expected, the spheroid compaction was reduced without acto-myosin activity (Fig. 3b), but not eliminated completely, suggesting that spheroid compaction is driven by multiple cellular mechanisms, one of which is acto-myosin contractility. Also as expected in spheroids that do not contract fully, the MSGs read out reduced levels of stress under these conditions (Fig. 4). The results and discussion section describing these results has been updated accordingly, and the relevant updated datasets are reproduced in this response document.

Figure 3b (updated). Quantification of spheroid size. Substantial compaction occurs over the first 24 hours, and is significantly reduced in MCS when actomyosin contractility is inhibited with blebbistatin (~50 % in MCS size). Data reported as mean \pm standard deviation, $n = 9$, * indicates $p < 0.05$ (one-way ANOVA with Tukey post-hoc pairwise comparisons).

Figure 4 (updated). Spatial mapping of cell-generated stresses within 3D fibroblast spheroid cultures. (a-d) Internal tissue stresses in the (a) radial and (b) circumferential direction are measured at days 0 ($n = 9$), 1 ($n = 17$), and 2 ($n = 35$) of culture. By day 2, compressive stresses in the radial and circumferential directions as large as 1.3 kPa are measured within the spheroid, while (inset) tensional stresses are measured in the circumferential direction at the outer surface of the spheroid. Internal tissue stresses in the (c) radial and (d) circumferential direction are greatly reduced in MCS cultured with blebbistatin to inhibit actomyosin contractility. Red data points represent tensional stress measurements, blue data points represent compressional stress measurements, and black data points represent stress measurements close to zero (-10 Pa to +10 Pa). Error bars indicate a combination of the 95% confidence interval for each data point, based on a Monte Carlo simulation of stress measurement arising from precision-related errors in identifying the MSG edge and accuracy errors in defining MSG shear modulus. (e, f) The average stresses measured throughout the MCS in the (e) radial and (f)

circumferential directions increased significantly from day 0 to days 1 and 2, and was significantly reduced by inhibition of actomyosin contractility. * indicates $p < 0.05$ (one-way ANOVA with Tukey post-hoc pairwise comparisons).

2b- *In addition, cells seem to orient along a circumferential axis at the surface of the aggregate. That would be in line with the authors hypothesis. Can the authors measure cell circumferential alignment along the radial axis of the aggregate?*

The reviewer is correct. By eye, the cells do seem to orient circumferentially. To test this more formally, we measured (1) the elongation length of the nuclei along the major elliptical axis; (2) the aspect ratio (length/width) of the nuclei, and (3) the relative orientation of cell nuclei to the tissue circumference in our tissue sections; as a function of radial position within the spheroid. The following datasets represent nuclear morphology information for all the cells in a tissue section, repeated over three separate spheroids, and oriented in reference to the radial position of the cell within the spheroid.

Figure not included in manuscript. Nuclear morphology within MCS tissue sections as a function of position within the spheroid, using several metrics. There is a slight trend towards nuclear elongation towards the edge of MCS as quantified by (a) circumferential elongation and (b) aspect ratio, but indicate no trends towards nuclear alignment at the MCS edge as (c) the degree offset from the edge is scattered evenly throughout the spheroid.

Nuclear orientation angle does not correlate with radial position, and while there is a trend towards increasing nuclear elongation towards the outside edge of the spheroid, this is only clear for a relatively small cell subpopulation. Hence, statistical analyses revealed no significant differences. We believe these surprising quantification results arise because the 2D tissue sections of the spheroids used here (which are necessary to allow high-resolution imaging deep within the dense tissue) do not capture nuclear orientation “into the page”. Since nuclei will orient along this axis even on the tissue periphery, we believe this may explain why the quantified data does not match our initial impression by eye.

Since the hypothesis that spheroid compaction drives the observed internal stress profiles has already been supported with the blebbistatin experiments outlined in Reviewer 1, point 1, as well as the previous observations for F-actin, p-myosin (Fig. 5f-i), and the distribution of cell density through the spheroid (Fig. 6b); we decided not to include these potentially confusing nuclear morphology results in the manuscript, but would be happy to include it as supplemental information if the reviewer feels that it adds value to the work.

3-The authors exclude cell proliferation as a source of tissue stress. This is based on the distribution of proliferation pattern. Can the authors interfere with cell division (using for example inhibitors of replication) to actually test this hypothesis?

We used mitomycin-C to block proliferation (data presented below) in the spheroid, and observed that this causes only a very small change (~10 %) in the average compacted spheroid size (see attached figure; data reported as mean \pm standard deviation, n = 13, * indicates p < 0.05; error bars too small to see). Hence, while proliferation may be responsible for generating some small baseline forces within the compacting tissue, this contribution will likely be < 10% of the current stress readings. Given the spread of data values for stresses observed within a spheroid, any differences will not be detectable when comparing average internal stresses between spheroids, and an experiment testing stresses while blocking proliferation would be inconclusive.

Figure not included in manuscript. Quantification of spheroid size when cell proliferation is inhibited by mitomycin c. No differences in spheroid size occurs until the second day of culture and causes only a very small change (~10 %) in MCS size. Data reported as mean \pm standard deviation, n = 13, * indicates p < 0.05; error bars too small to see in print.

More importantly however, the conclusion that we draw was not that proliferation be excluded entirely as a source of stress, but that proliferation be excluded as a source of differential stress patterns that could arise within the spheroid. If proliferation occurs uniformly through the spheroid, no stress profile is generated (Fig. S17), and our analysis of proliferation patterns (Fig. S18) reveals no spatial patterns that would contribute towards the overall stress pattern observed. Hence, blocking proliferation may change overall stress levels in the spheroid, but will not provide further support for this specific hypothesis. The section “Biomarker analysis is inconsistent with growth...” has been edited to clarify this important distinction, and now clearly states that “Since uniform proliferation in an unbounded MCS causes no additional stresses (Fig. S18), we reasoned that stress patterns can only arise if spatially distinct proliferation patterns occur.”

4-In Figure 3, the authors show the normalized diameter of the aggregates over time. The normalization prevents the reader from knowing the actual size, and its variation, of the aggregates. Can the authors show the actual size with an estimation of the variability (mean \pm standard deviation)? This is very important as the authors go on to categorize regions of the aggregates as “the core region at the center of the spheroid, the intermediate zone (50 to 150 μm below the surface), and the edge (0 to 50 μm below the surface)”. It is difficult to evaluate this categorization without the actual size measurement.

We have updated Fig. 3b with actual spheroid sizes during the formation process. At day 2, the average diameter of spheroids tested without blebbistatin inhibition was 800 μm . The data, presented for $n = 9$ spheroids, demonstrates very low variability in the initial and final spheroid sizes in all experiments. In addition, we have also updated Fig. 4 with dual x-axis scales that allow easy understanding of the zone sizes.

Figure 3b (updated). Quantification of spheroid size. Substantial compaction occurs over the first 24 hours, and is significantly reduced in MCS when actomyosin contractility is inhibited with blebbistatin (~50 % in MCS size). Data reported as mean \pm standard deviation, $n = 9$, * indicates $p < 0.05$ (one-way ANOVA with Tukey post-hoc pairwise comparisons).

5a-The authors describe briefly how the addition of the collagen layer on top of the MSG could affect the measurement. Actual measurement of the thickness and uniformity would be good to have. Also, they estimate the elastic modulus of the collagen layer but I could not find how this is calculated. Finally, to actually know how this affects the measurement, varying the thickness of the collagen would be a good test of their system.

We understand that the reviewer is requesting thickness and uniformity measurements of the collagen coating, to further confirm that the collagen coating does not affect the MSG mechanics. We agree that our previous analysis was unsatisfying because we could not experimentally measure the mechanical stiffness of the collagen film on the ultrasoft and topographically complex MSG surface, and relied on using values from literature. Furthermore, the mechanics of collagen coatings are notoriously variable based on small changes in environmental parameters (temperature, humidity, etc). Hence, rather than attempt to tease apart the mechanics of the surrounding collagen layer using a reductive approach, we believe that this issue can be directly and better assessed by measuring the mechanical rigidity of individual MSGs, to capture all sources of experimental variability in a single experiment.

We implemented an osmotic pressure measurement technique to determine the mechanics of individual MSGs. Briefly, an osmotic pressure is applied to the MSG using a long-chain dextran polymer that cannot enter the MSG polymer matrix. Gels are compacted when placed in this solution, and the degree of deformation in response to osmotic pressure can be used to determine rigidity of the MSG (detailed description and references provided in supplemental methods). Using this technique, we compared the mechanical stiffness of uncoated and collagen-coated MSGs, and our results demonstrate that there is no measurable difference in mechanical properties between coated and uncoated MSGs (data presented in Fig. S10, detailed description of technique in Supplemental Methods, and results and discussion have been updated in the main manuscript).

Figure S10 (updated). Measurement of MSG mechanical properties by application of osmotic pressure. An aqueous solution of long-chain dextran (500 kDa) was used to exert an osmotic pressure on polyacrylamide hydrogels. The dextran polymer chains are too large to enter the polyacrylamide pores, and are therefore excluded from the hydrogel. The osmotic pressure differential forces water out of the MSG, which deforms in proportion to the MSG mechanical compliance. (a) A schematic representation and (b) fluorescent microscope images (scale bar = 50 μm) depicting hydrogel contraction when exposed to 100 mg/mL of dextran solution. (c) MSG sizes remain constant after 3 hours in the dextran solution, confirming that dextran chains are excluded from the polymer matrix ($n = 19$). The system was calibrated against osmotic pressure-induced deformation of a bulk disk-shaped polyacrylamide hydrogel sample (diameter = 13 mm) for which the shear modulus was established using conventional shear rheometry. (d) A finite element simulation was developed to determine the effective

osmotic pressure generated by a 100 mg/mL solution of dextran. The parametric sweep of external pressures on samples was used to determine that 100 mg/mL of dextran exerts 67 Pa pressure on the hydrogel surface. Next, this osmotic pressure value was applied to (e) a parametric sweep of shear modulus in the isotropic compression of a spherical MSG. (f) Osmotic pressure measurements on MSGs indicates that collagen coating does not significantly alter mechanical rigidity of the MSG ($n = 24$, $p = 0.782$). (g) No significant differences were found between coated MSGs (control) and MSGs that had been removed from spheroids after two days of culture by detergent-based extraction (released), demonstrating that MSG properties remain constant even after embedding within the tissue of interest ($n = 16-19$, $p = 0.837$). All data reported as mean standard deviation. NS indicates no significant differences (one-way ANOVA with Tukey post-hoc pairwise comparisons).

5b-*Also, on page 6, authors claim that “mechanical properties of the MSGs can be considered constant even after embedding in the tissue of interest”. However, the controls on collagen distribution appear to have been done on MSGs outside of tissues (Figure 1). If that’s right, the claim should be changed.*

We have now conducted new experiments that do allow us to conclude that MSG mechanical properties can be considered constant even after embedding in the tissue of interest. We released MSGs from spheroids by detergent-based lysis with SDS and measured their mechanical rigidity using the osmotic pressure technique previously mentioned. We confirmed that MSGs return to their original size (Fig. 4; day 0 stresses), and that osmotic pressure testing (Fig. S10g) reveals no significant change in mechanical rigidity. Taken together with the images previously presented in Fig. 1e which demonstrate no incursion of collagen into the MSG body, we can now conclude that mechanical properties of MSGs remain constant even after embedding. These new experiments replace the previous Fig. S9 (that reported bulk gel rheometry values) with the current Fig. S10. The results and discussion section have been modified to reflect these findings. Please refer to the response to the previous question for an updated figure.

6-*The authors explain how reducing the stiffness of MSGs increases the resolution of the measurement. Can they discuss how low in stiffness and resolution their system can go? What happens to the MSGs when they are too soft?*

There are several challenges in working with softer MSGs, including difficulty finding the gels in 3D tissues, due to a reduction in polymer sites to graft the fluorescein monomer, and difficulties in separating MSGs from each other once the surfactant has been removed from the system. In our hands, we were unable to produce MSGs softer than $G = 60$ Pa, using the described methods.

A brief discussion on these limitations has been included in the Discussion section of the manuscript: “Further reductions in polymer content to produce softer MSGs failed because MSGs self-aggregated when the surfactant was removed from the system, and the reduced fluorescent content made them difficult to find in 3D tissues.”

7-The authors describe the limitations of previously developed methods to measure the stress applied by tissues. More details on the limitations of this new technique would be fair. What is the range of stress that can be measured? What types of tissues can it be used with? Can other ECM components be used? The authors bring forward the advantage of being able to measure isotropic stress but make little use of this measurement. What context would that be useful for?

We have developed a new discussion section, in which we specifically discuss the fundamental limitations in using this general strategy, as well as the practical limitations of the specific MSGs used in this work. We respectfully disagree with the statement that we make little use of the ability to measure isotropic stress components in these tissues. Reporting absolute values for radial and circumferential (orthogonal) stress magnitudes are a direct consequence of being able to measure both isotropic and anisotropic stress components. Techniques like the oil-droplet method can only measure anisotropic stresses, and the isotropic stress that contributes to both radial and circumferential stress magnitudes cannot be measured. To address this comment, the discussion has been updated as follows:

The soft MSGs fabricated allow stress measurements with as little as 10 Pa resolution, and a range of > 100 Pa (tension) to several kilopascals (compression)....

Some fundamental limitations must be considered in the use of MSGs. First, because MSGs are porous hydrogel structures, they will not deform in response to altered hydrostatic fluid pressure, and can only be used to measure solid mechanical stress. Second, the viscous nature of the biological tissue being studied must be carefully considered. Most biological tissues exhibit some form of viscoelastic behaviour, and behave as elastic solids on short timescales and as fluids on long time scales⁵⁶⁻⁵⁸. If the tissue behaves like an elastic solid, then the MSG within the tissue can be considered as an inclusion within an elastic matrix. In such cases, the modulus of the surrounding tissue and the contact boundary conditions at the MSG-spheroid interface plays a critical role in deformation of the MSG⁵⁹. Hence, the presented analysis applies only if the tissue exhibits fluid behaviour to dissipate inclusion-generated stresses. In the case of MCS in which cells form transient adhesions and can migrate over each other, the time constant of solid-liquid transitions has previously been established to be on the order of minutes⁶⁰. Although tissue components such as secreted extracellular matrix proteins could slow cell movement, we confirmed that ECM in our spheroids is not in mature fibrillar form (Fig. S13b) and is hence unlikely to stabilize the fluid behaviour beyond this time frame. Hence, measurements taken in spheroids at 24 hour intervals will not be affected by this factor.

Aside from these constraints, we anticipate broad utility of the MSGs in furthering our understanding of the relationship between mechanical forces and biological function, as this technique can readily be extended to study a wide variety of 3D tissues. First, although we have limited our studies to coating MSGs with one ECM protein, the ability to apply the well-established sulfo-SANPAH chemistry to incorporate a wide variety of proteins and adhesive motifs allows the MSG to be tailored for specific extracellular matrix contexts without disrupting native tissue function. Similarly, mechanical rigidity of the polymer formulation achieved here is comparable to the softest body tissues⁶¹, and can be readily stiffened by increasing the crosslinker density. Hence, MSGs can be designed to closely match the native composition and mechanics of the tissue of interest, to 'mask' the presence of the MSG and prevent the formation of a fibrotic capsule or other immune rejection response. Second, the need to obtain the zero-stress state dimensions of the MSG limits workflow. Generating monodisperse hydrogel sensor size using microfluidic production strategies would eliminate the need to sacrifice the tissue for MSG analysis. Third, the current MSG technology requires analysis by optical means. This is challenging in dense tissue cultures, and we were unable to make reliable measurements more than 300 μm beneath the tissue surface, nor were we able to resolve 3D deformations at this distance. Advanced imaging techniques such as using optical windows for in vivo imaging; or excising, fixing and clarifying tissues may be suitable. Alternatively, the MSG

sensors may be decorated with brighter quantum dot-based labels, X-ray contrast agents, or acoustic contrast agents for analysis with different imaging modalities. Finally, the analysis technique developed here considers only ellipsoidal geometries for MSG deformation, and explicitly assumes that stresses within an MSG are uniform and have at least one axis of symmetry. While this is justified for analysis of supra-cellular stresses in MCS cultures, this approach does not capture sub-cellular stresses that must be present, and would not be sufficient to analyze deformation in a non-symmetric tissue. Fortunately, computational approaches to model mechanical stresses in each individual MSG can be readily developed⁴¹.

8-In Figure 3c, the authors show 3 inserts with MSGs at different locations of the aggregates. This gives the impression that multiple MSGs are placed within one aggregate, which is not the actual experiment, as far as I understood. This is giving the reader the wrong idea. Can the authors change to show 3 actual aggregates with the MSGs as an overlaid image? This will show the MSGs in their actual context. Moreover, a high-resolution image of the cells around the MSG would be very good to evaluate how cells interact with the MSGs.

We apologize for the confusing image and layout. We have updated this figure with three low magnification images of aggregates, paired with high-magnification images of MSGs, showing the various orientations of embedded MSGs (Fig. 3e). We have also included a high resolution confocal image of an MSG partially embedded within an MCS (Fig. 3d; cells labelled for F-actin and nuclei with phalloidin red and Hoechst nuclear stain), which shows interaction between cells and the MSG.

Figure 3d, e (updated). (d) 3D confocal reconstruction of MSG (green) embedded at the edge of a spheroid at day 2 of culture (scale bar = 50 m). (e) Embedded fluorescent sensors deform within the spheroid, with circumferential, radial, or no orientation (scale bar = 50 m), based on position within the spheroid.

10-Images of the aggregates in Figures 4 and 5 show many gaps in the tissue, in particular at the core. Is that correct? How much fluid is there? Is this fluid under high pressure? Is that fluid also exerting pressure on the MSGs?

We believe that there is more extracellular space towards the core of the spheroid based on the histology sections of spheroids. This space is likely filled with nutrient media, given the way the

spheroid is formed by aggregation of floating cells. This fluid may be under pressure depending on the porosity of the surrounding tissue, but since fluid pressure is an isotropic force (acting equally in all directions), it does not contribute to additional deformation of the porous MSG. In hydrogels, internal fluid pressure will rise with the application of external fluid pressure, creating no net deformation across a porous boundary. Hence, only “solid” stresses can be measured with this technique, and this important point has been included in the discussion section as part of fundamental limitations of using MSGs, by adding the following line:

“Some fundamental limitations must be considered in the use of MSGs. First, because MSGs are porous hydrogel structures, they will not deform in response to altered hydrostatic fluid pressure, and can only be used to measure solid mechanical stress.”

11-On page 7, “HS-5 MCS cultures compacted dramatically over the first 24 hours”. Maybe “dramatically” is exaggerated.

Thank you, this has been corrected.

Reviewer #2 (Remarks to the Author):

1. The polyacrylamide gel particles exhibit an elastic modulus of 100 Pa, which is far below the typical range used for PA gels. This would suggest that the particles might be highly viscous. While the loss modulus is reported at 10 Hz, a frequency sweep for the storage and loss modulus should be reported. If the loss modulus: storage modulus ratio is much higher at low frequencies (0.001 Hz), that could impact interpretation. This is important to understanding the capabilities of these stress sensors.

We agree with this important point and have conducted a frequency sweep analysis of the storage and loss modulus in “bulk” ultrasoft polyacrylamide gel formulations using a shear rheometer. The results are presented in Supplemental Fig. S8. The data demonstrates minimal changes in ratio between storage and loss modulus at frequencies below 1 Hz, and hence is unlikely to impact data interpretation. This important point has been included as a supplemental figure (Fig. S8) and as part of the “MSG mechanical characterization...” section of the revised manuscript.

Figure S8 (updated). Shear rheometry characterization of bulk, fluorescently-labelled polyacrylamide hydrogels. These bulk measurements demonstrate linear elastic material properties with negligible loss modulus (a) over large strain range (data repeated in main manuscript Fig. 2a) and (b) for load frequencies less than 1 Hz. For frequencies greater than 1 Hz, the reduced shear modulus is likely due to slippage of the hydrogel on the rheometer plates. However, frequencies greater than 1 Hz are unlikely to be relevant to the current application of MSGs in multicellular spheroids.

2. The mechanical properties reported appear to be based on bulk rheological measurements. This should be complemented by AFM measurements of the bead elastic modulus to confirm that beads (or another micro-scale technique), formed through a distinct process relative to the bulk gel, exhibit similar properties to the bulk gel.

We agree that a complementary method for mechanical testing is required to confirm the mechanical properties of the MSGs. However, AFM is particularly challenging to implement with ultrasoft spherical

gels of these sizes (10s of microns), because the deformations observed when embedded in the tissue are global, while standard AFM is best suited for higher spatial resolution measurements (microns) that do not interrogate mechanical properties deep into the material. Even if the AFM tip was modified with a platen, analysis of deformation of the spherical structure without detailed knowledge of the contact boundary conditions (area and slip coefficient) would introduce significant error to the measurement.

Instead, we used an osmotic pressure measurement technique to apply spherical loads and determine the global stiffness of individual MSGs. Briefly, an osmotic pressure is applied to the MSG using a long-chain dextran polymer that cannot enter the MSG polymer matrix. Gels compact when placed in this solution, and the degree of deformation in response to polymer concentration can be used to determine rigidity of the MSG (detailed description and references to literature provided in supplemental methods). Using this technique, we characterized the mechanical stiffness of individual MSGs as a function of MSG size (Fig. 2c), with and without collagen coatings (Fig. S10f), and before and after embedding and release from a spheroid (Fig. S10g). These results demonstrate minimal variation in mechanical stiffness between bulk rheology and microscale MSG characterization. A discussion on these findings have been included in the “MSG mechanical characterization...” section of the manuscript.

Figure S10 (updated). Measurement of MSG mechanical properties by application of osmotic pressure. An aqueous solution of long-chain dextran (500 kDa) was used to exert an osmotic pressure on polyacrylamide hydrogels. The dextran polymer chains are too large to enter the polyacrylamide pores, and are therefore

excluded from the hydrogel. The osmotic pressure differential forces water out of the MSG, which deforms in proportion to the MSG mechanical compliance. (a) A schematic representation and (b) fluorescent microscope images (scale bar = 50 μ m) depicting hydrogel contraction when exposed to 100 mg/mL of dextran solution. (c) MSG sizes remain constant after 3 hours in the dextran solution, confirming that dextran chains are excluded from the polymer matrix ($n = 19$). The system was calibrated against osmotic pressure-induced deformation of a bulk disk-shaped polyacrylamide hydrogel sample (diameter = 13 mm) for which the shear modulus was established using conventional shear rheometry. (d) A finite element simulation was developed to determine the effective osmotic pressure generated by a 100 mg/mL solution of dextran. The parametric sweep of external pressures on samples was used to determine that 100 mg/mL of dextran exerts 67 Pa pressure on the hydrogel surface. Next, this osmotic pressure value was applied to (e) a parametric sweep of shear modulus in the isotropic compression of a spherical MSG. (f) Osmotic pressure measurements on MSGs indicates that collagen coating does not significantly alter mechanical rigidity of the MSG ($n = 24$, $p = 0.782$). (g) No significant differences were found between coated MSGs (control) and MSGs that had been removed from spheroids after two days of culture by detergent-based extraction (released), demonstrating that MSG properties remain constant even after embedding within the tissue of interest ($n = 19-16$, $p = 0.837$). All data reported as mean \pm standard deviation. NS indicates no significant differences (one-way ANOVA with Tukey post-hoc pairwise comparisons).

3. The MSG sensors are coated with collagen. This is an unusual choice, as cells actively sense and respond to collagen-coated hydrogels (i.e. exert traction forces on the gels in a stiffness dependent manner). As such, the beads may induce cells to actively deform them and thus the measurements are not passive readouts of stress in the spheroid. The authors should also conduct and report measurements with uncoated gels, Given that the particles are embedded in the multicellular spheroids, this should provide a passive readout of local stresses. In addition, or alternatively, the authors could probe stresses measured when the gauges are coated with a cell-cell adhesion ligand (i.e the n-cadherin mimetic ligand like HAVDI – see Cosgrove, et al., Nature Materials 2016) as a better mimic of the stresses that a cell would experience in a spheroid.

The reviewer raises an excellent point, and we consider the well-established surface functionalization techniques to coat polyacrylamide with a wide variety of adhesive ligands to be a strong positive aspect in improving applicability of this system towards many tissues. We agree that the choice of adhesive ligand could play a critical role in activating local cell response. To address this, we first attempted to carry out the suggested experiment and cultured uncoated MSGs within spheroids. We found that uncoated MSGs are incorporated into spheroids at extremely low rates during the tissue formation process, and most of these are then “ejected” from the spheroids by day 2 of culture. Alternatives such as using the the HAVDI ligand would be an excellent strategy to reduce the contractile state of cells (as shown in MSCs by Cosgrove et al. within the spheroid, and would be an interesting next step. However, using this ligand to reduce the contractility of cells around the MSG may also alter the “native” baseline contractility of cells within the spheroids, particularly at the sites of stress measurement.

Hence, to select the most appropriate and minimally disruptive coating ligands, we believe that an analysis of ECM native to the spheroid would be most helpful. We originally chose type I collagen

because HS-5 cells in 2D culture are known to secrete type I collagen (Roecklein et al., PMID 7849321). To confirm whether this is true in spheroid cultures, we immunostained tissue sections for de novo production of type I collagen, and confirmed that this ECM is natively present in the spheroid after two days in culture. Hence, coating the MSGs with type I collagen is unlikely to provoke any non-native contractile behaviour. This important justification, data and discussion have been included in:

- “Distinct spatial patterns...” results section:
 “Since the HS-5 MCS cultures secrete Type I collagen (Fig. 3c; Fig. S13) [Ref 51], coating the MSGs with this extracellular matrix protein is unlikely to alter baseline cell or tissue contractility”
- Figures 3c and S13a.

Figure S13a (updated; second panel also reproduced as Fig. 3c). Characterization of MCS structure 48 hours after formation. (a) Type I collagen fluorescent immunostaining on sectioned MCS confirms that HS-5 fibroblasts secrete Type I collagen over 2 days of culture (scale bar = 250 mm). Negative control performed without primary antibody confirms that the signal detected is not a result of non-specific binding.

4. The data in Figure 3d and 3e, upon which the conclusions about the stress profile are based, are not very convincing. There is a large scatter in the data, and the data are fit with some hand-drawn dashed line. A few suggestions here:

4a. first, the authors should normalize the x-axis to be the distance between the edge (0) and the center of the spheroid (1). This makes the results easier to interpret as the beads are likely of different size.

We agree and have provided a second normalized x-axis to aid in interpretation.

4b. Second, the authors likely need a much higher number of measurements here to have any confidence in the stress profile, given the scatter (for example in 3e at 0, one of the triangles is measured at 0 and the other is at 115 Pa; and there appears to be only one data point near the center of the spheroid).

Re: Measurements at the spheroid surface

We have obtained more data points close to the edge of the spheroid, and consistently show that the outer 30 microns of the spheroid are under tension (updated Fig. 4b). Finding MSGs in this region is challenging, because the surface layer under tension is only a few cells thick, and if the MSG “breaches” the surface it is quickly ejected. We do however consistently observe this tension effect in all our experiments.

Re: Measurements in the spheroid core

[Note: similar response to Reviewer 1, point 1; Reviewer 2, point 4b; and Reviewer 4, point 11.]

We agree that the single point is not sufficient. To add more data, we attempted additional experiments, and although we improved our sample size (Fig. 3f; Fig. 4), we could not image MSGs deeper than 300 μm below the MCS surface, likely due to the optical density of the spheroids. We attempted two-photon confocal microscopy but were unable to penetrate further without unacceptable signal losses. Hence, we removed the single data point as it could not be repeated at this depth. Our additional data indicates that there is a small decrease in the average stress in the core region compared to the intermediate zone, but this difference is not statistically significant ($p = 0.24$ by two-tailed ANOVA), likely due to the high spread in measurement values. Hence, this does not allow us to conclude that the stress plateaus. We have therefore restated this observation, incorporated our new data and analysis, and explicitly acknowledge the limitation of this fluorescent optical imaging modality in the discussion section of the manuscript. The relevant updated datasets and analysis are reproduced in this response.

Selected panels from Figure 4 and Figure 5 (updated). Spatial mapping of cell-generated stresses within 3D fibroblast spheroid cultures, demonstrating additional data points collected within the spheroid. Internal tissue stresses in the (a) radial and (b) circumferential direction are measured at days 2 of culture (Figure 4). Using the

updated analysis models developed in response to reviewer comments, compressive stresses in the radial and circumferential directions as large as 1.3 kPa are measured within the spheroid, while (inset) tensional stresses are measured in the circumferential direction at the outer surface of the spheroid. In Figure 5 (d, e) stress measurements are grouped based on location for statistical comparison. * indicates $p < 0.05$ (one-way ANOVA with Tukey post-hoc comparisons).

4c. Third, the authors need to figure out the appropriate way to support their conclusions with statistical tests – perhaps a comparison with the day 0 profile could be instructive here?

We analyzed MSG stress profiles at Day 0, 1 and 2, with and without inhibition with blebbistatin; and presented the results in Fig. 4. A statistical analysis of the new data sets is provided in in Fig. 4e (radial) and Fig. 4f (circumferential), in which the average stress magnitudes throughout the spheroid is calculated. Average stress magnitudes increased significantly between days 0 and 1 ($p < 0.002$), and between days 0 and 2 ($p < 0.001$; one-way ANOVA with Tukey post-hoc comparisons), and was significantly reduced on days 1 and 2 when treated with 50 μM blebbistatin during the spheroid formation process ($p < 0.05$). In addition we have grouped radial and circumferential stresses at day 2 based on their locations within the spheroids, and conducted statistical tests to confirm that the edge of the spheroid has markedly distinct forces than the intermediate and core (Fig. 5d, e).

Figure 4 e, f (updated). Spatial mapping of cell-generated stresses within 3D fibroblast spheroid cultures. (e, f) The average stresses measured throughout the MCS in the (e) radial and (f) circumferential directions increased significantly from day 0 to days 1 and 2, and was significantly reduced by inhibition of actomyosin contractility. * indicates $p < 0.05$ (one-way ANOVA with Tukey post-hoc pairwise comparisons).

Figure 5d, e (updated). Characterization of biological markers for cell mechanical activity within the MCS. (d, e) Quantification of average (d) radial and (e) circumferential stresses within the three zones of MCS on the second day of culture: edge ($n = 5$), intermediate zone ($n = 15$), and core ($n = 15$). * indicates $p < 0.05$ (one-way ANOVA with Tukey post-hoc pairwise comparisons).

4d. Finally, the authors should include the raw data for bead extension and orientation vs position in the sphere that 3d and 3 are based on.

We have included the requested raw data as supplemental data tables (Table S1-S4)

5a. The process studied here – of multicellular spheroids formed and then allowed to contract over 3 days is distinct from the often-studied case of spheroids allowed to proliferate from a single cell starting point (i.e. Helmlinger, et al., Nature Biotechnology, 1998). What the authors are studying is a contractile process as cells contract as they form cell-cell junctions, and not a spheroid expansion process due to proliferation (there is probably not much cell division over 3 days). While there is no problem with this approach, the authors should be careful about making broad conclusions. For example, there is no reason that it would be expected that “compression [would be] induced by proliferative growth” as written in the abstract as the spheroid is still in the contractile phase. Similarly, the statements, “did not support the conclusion that tension at the edge is caused by proliferative expansion” and “Our results confirm, for the first time by direct experimental measurement, computational simulations that predict high tensions at the edge of tissues on patterned substrates of varying geometries⁵¹ and within growing tumours^{42,48}, as well as other indirect measurements of internal stress such as the degree of tissue relaxation following release of internal solid stresses⁴¹” should be qualified, as these contracting spheroids (which decrease in size) are likely to be different than growing tumors.

We agree with these statements and have qualified our claims accordingly. Specifically, we have

- 1) Rewritten the abstract, using the following:

These stresses develop through cell-driven mechanical compaction at the tissue periphery, and suggest that the tissue formation process plays a critically important, but previously unrecognized role in specifying mechanobiological function.

- 2) Rewritten the section on “Biomarker analysis is inconsistent with growth...” to clarify that we are testing whether internal stress may be caused by patterns of proliferation... as this occurs in tissues grown from a single cell for which similar stress profiles are seen.

- 3) Rewritten the discussion section to make our summaries more precise, as follows:

Interestingly, these results indicate that similar stress patterns in aggregated MCS and those grown from single cells are generated by two distinct mechanisms, suggesting that (1) MCS formed by aggregation may be mechanobiologically comparable to tumours grown from a single cell; and (2) the tissue formation process plays an important role in regulating the mechanical state of the tissue. These findings may explain why spheroids formed using different methods could result in distinct biological functions and be difficult to compare across research labs, as described by others⁵⁵. Hence, the spheroid formation process may be more important than previously considered, when trying to construct tissues that have mechanobiological sensitivities or functions.

6. The YAP/Taz result is potentially interesting, however, the stains are somewhat difficult to see and interpret. The standard in the field is to use immunofluorescence and to quantify the ratio of nuclear:cytoskeletal YAP. The authors should do this.

We quantified the nuclear:cytoskeletal YAP ratios using software (Aperion) that enables this analysis for samples labelled by IHC. This method has previously been validated by us against immunofluorescent techniques (Knight et al., 2018 Cell Reports, DOI: 10.1016/j.celrep.2018.02.095). We found that IF antibody-based stains for YAP/TAZ worked well for cells cultured on 2D surfaces, but were less robust in our hands when staining fixed tissue sections. This updated analysis, now shown in Fig. 5j, k in the manuscript, confirms that nuclear localization of YAP is significantly increased in the edge region of the spheroids.

Figure 5j, k (updated). (j) Immunohistochemistry for YAP/TAZ on sectioned spheroids shows significantly greater nuclear abundance in the periphery when compared to the core, indicating that the observed forces have biological significance in their ability to activate key signalling pathways (scale bar = 50 μ m). (k) YAP/TAZ nuclear intensity was classified as either predominantly nuclear, predominantly cytoplasmic, or nuclear = cytoplasmic based on its relative abundance. There was a significant increase in the fraction of cells with nuclear > cytoplasmic YAP at the edge of the spheroid when compared to both the intermediate zone and the core. Data reported as mean standard deviation, n = 13, * indicates p < 0.05 when compared to similar threshold levels at other spatial locations (one-way ANOVA with Tukey post-hoc pairwise comparisons).

7. The comment “Consistent with several studies^{51,52}, these results suggest that the spatial patterns of endogenous forces that arise within 3D clusters of aggregated cells could play an important role in specifying stem cell lineage commitment” is highly speculative, and should be moved to discussion section.

We agree, and have moved this comment to the discussion section

Reviewer #3 (Remarks to the Author):

1. While this is a very interesting strategy for measuring local stress in a 3-D living environment, there is a major concern about how accurate estimates of the local stress are for the following reasons. It is well known that the stress field surrounding inclusion in an elastic matrix depend on many factors, including the mismatch of the elastic moduli of the matrix and inclusion, the location of the inclusion, the boundary conditions at the inclusion-matrix interface (perfect bonding or sliding) (cf. Jasluk et al. J Appl Mech 1997).

1a. Since the stiffness of the embedded spheres does not match the stiffness of the surrounding multicellular spheroid the estimates of stress in the spheroid may not be accurate. This mismatch is further accentuated by the finding that there is a stiffness gradient in the radial direction of the spheroid.

1b. Furthermore, it is not clear what are boundary conditions at the sphere-spheroid interface; is it a firm bonding or there is sliding, which also bears effect on the stress field surrounding the sphere. If I understood correctly, the authors argued that the above concerns are valid only for inert matter: "For static structures in which MS is rigidly coupled to the surrounding matrix, and the matrix does not have the opportunity or capacity to reform and remodel, MSG readouts will be coupled to the stiffness of the surrounding tissue." (l. 287-289). However, since the spheroid is a living matter "in which cells are free to undergo plastic reorganization through morphological adaptation and migration during spheroid formation" (l. 286-296) stiffness mismatch between the spheres is not an issue. I am not convinced that this is true. Some stronger argument/evidence is needed.

We agree with these points and acknowledge that (1a) modulus mismatch and (1b) boundary contact conditions can influence accuracy of stress measurements, when the MSG is modelled as an inclusion within a deforming elastic matrix. As alluded to, but inadequately supported and explained in our original manuscript, viscous flow of the tissue around the MSG is distinct from this situation, and we were previously incorrect to model the system as an inclusion. The premise of viscous flow in biological tissues is not unusual and implicit in all traction force microscopy analysis techniques developed in this field, in which the modulus of the biological cell/tissue, and the nature of the boundary between the tissue and the underlying polyacrylamide substrate is not relevant to the analysis, because cells 'flow' by attaching, detaching and reattaching on the elastic adhesive substrate. These arguments are presented and supported in detail as follows:

Validity of the viscous flow assumption

The assumption that spheroids behave as a viscous fluid is well supported by the body of literature that considers biological tissues to be viscoelastic, and behave as elastic solids on short timescales and as fluids on long time scales (Schötz E, et al., HFSP journal DOI:10.2976/1.2834817; Schotz et al., J. Royal Soc. Interface 2013, DOI: 10.1098/rsif.2013.0726; Bi et al., Nature Physics 2015, DOI: 10.1038/nphys3471). Cell aggregates in particular can be considered as viscous fluids because:

- (1) the components of a cell aggregate (cells) can slip over each other. For illustrative purposes, an analogy would be cornstarch in water under external load, where the particles demonstrate solid-like behaviour by "jamming" at short time scales, but behave as a liquid and slip over each other at longer time scales (Waitukatis et al., Nature 2012, DOI: 10.1038/nature11187).

- (2) Unlike cornstarch, cells actively adhere onto surfaces through integrins, but these adhesions are transient. This is evidenced by a cells ability to migrate while adhered on a substrate. Many cells including the fibroblasts used in this work are motile, and will actively crawl on a substrate in response to applied shear stresses as low as 2 Pa (Gaanich et al., DOI: 10.1152/ajpheart.00578.2006).
- (3) The time constant of solid-liquid transition in cell aggregates has previously been established to be on the order of minutes (Delarue et al., Interface Focus 2014 DOI: 10.1098/rsfs.2014.0033). Although tissue components such as secreted extracellular matrix proteins could slow cell movement, we confirmed that ECM in our spheroids is not in mature fibrillar form (Fig. S13b) and hence unlikely to stabilize the fluid behaviour.

Hence, it is quite reasonable to assume that spheroids behave as viscous fluids around the MSG at the time scales (48 hour periods) presented in this work.

Consequences of the viscous flow assumption

If the premise that spheroids behave as viscous fluids is accepted, then the well-established behaviour of inclusions in an elastic matrix does not apply, and we were previously incorrect to model the system this way. Flow in the biological tissue will dissipate any stresses that arise due to the inclusion (either by stiffness mismatch or boundary contact), and failing to account for this factor leads to inaccurate stress profiles in the surrounding tissue. To incorporate this important parameter into our revised simulations, we consider the MSG alone under globally defined strain conditions, by specifying uniform axial and/or radial strain through the MSG body. We confirmed that this strain-controlled boundary condition results in uniform axial and radial stress patterns throughout the MSG, as expected when the tissue exhibits viscous flow (now included as Fig. S11). Hence, if the tissue flows around the MSG, the mechanics of the surrounding tissue is irrelevant. Since this paradigm is consistently applied in the traction force microscopy literature, which is a conceptually similar problem to the work being developed here, we believe that it is a valid approach to the present analysis, and issues such as stiffness mismatch and boundary contact conditions should no longer be considered.

Changes made to the manuscript

We have captured the key points of this analysis in the discussion section (fundamental limitations of MSGs), and have provided a detailed description of methods and results for the new simulation strategies (Supplemental Methods, “MSG mechanical characterization and analysis of deformation”). The results of the new simulations are presented in Fig. 2e, f; Supplemental Fig. S11 and S12. New simulations have been applied to all analyses presented in the revised manuscript (Fig. 4; Fig. 5). The previous figures discussing the role of stiffness mismatch have been removed, as these are incorrect.

Finally, a key limitation that has emerged from this updated analysis is that it is important to consider the solid-liquid time constant and hence the experimental sampling period when applying MSGs to study other tissues. For example, in cells stimulated to contract by addition of a chemical factor, MSG measurements taken seconds after the contractile event and before the biological material is allowed to remodel will not provide accurate stress readings. Hence, any results must be considered carefully in light of the viscous nature of the specific tissue under study. A brief description of this limitation has been included in the discussion portion of the manuscript, as follows:

Figure S11 (updated). Finite element model to simulate MSG deformation. (a) Schematics of the two-dimensional axisymmetric model along with strain conditions applied throughout the bead domain. (b) Representative image of a 100 μm MSG bead deforming under -0.33 axial strain and -0.5 radial strain domain conditions. Corresponding (c) axial and (d) radial stresses are confirmed to be uniform throughout the MSG, consistent with the concept of viscous flow in the surrounding tissue.

Relevant discussion section:

Second, the viscous nature of the biological tissue being studied must be carefully considered. Most biological tissues exhibit some form of viscoelastic behaviour, and behave as elastic solids on short timescales and as fluids on long time scales^{56–58}. If the tissue behaves like an elastic solid, then the MSG within the tissue can be considered as an inclusion within an elastic matrix. In such cases, the modulus of the surrounding tissue and the contact boundary conditions at the MSG-spheroid interface plays a critical role in deformation of the MSG⁵⁹. Hence, the presented analysis applies only if the tissue exhibits fluid behaviour to dissipate inclusion-generated stresses. For example, measurements taken seconds after cells in a collagen matrix are stimulated to contract by addition of a chemical factor would not allow the cells to flow and would therefore not provide accurate stress measurements. In the current case of a MCS in which cells form transient adhesions and can migrate over each other, the time constant of solid-liquid transitions has previously been established to be on the order of minutes⁶⁰. Although tissue components such as secreted extracellular matrix proteins could slow cell movement, we confirmed that ECM in our spheroids is not in mature fibrillar form (Fig. S13b) and is hence unlikely to stabilize the fluid behaviour beyond this time frame. Hence, measurements taken in spheroids at 24 hour intervals will not be affected by these factors. -----

Minor Comments:

2a. 116-120, Fig. 2a: To the first approximation, the shear modulus of the hydrogel appears to be constant over a broad range of strains (i.e., linear elasticity). However, a more detailed inspection of the data in Fig. 2a reveals that at low strains the modulus exhibits a short-range softening, followed by a long-range stiffening, indicative of non-linear stress-strain behavior, which is expected for compressible materials.

The small variations in stiffness measured on the shear rheometer are within expected experimental uncertainty for the analysis of such soft materials. Hence, in our view we remain comfortable with the assumption of linearity, given the very small deviations observed. For clarity, we have replaced Fig. 2a with a modulus-strain curve that better represents the majority of our measurements. For further clarity, a dataset of stress-strain curves has been included in this response and shows a linear relationship between stress and strain.

Figure 2a (updated). Shear rheometry on bulk, fluorescently-labelled polyacrylamide hydrogels indicate linear elastic deformations over large strains, with negligible loss modulus.

Figure not included in manuscript. Stress-strain curve of the fluorescent polyacrylamide bulk material obtained using a shear rheometer. Data in the manuscript is presented as modulus-strain curves (Fig. 2a).

2b. If the gel were incompressible (e.g., rubber-like), then constant shear modulus over a broad range of strains is a likely behavior.

The polyacrylamide material is compressible, as evidenced by our measurements of Poisson's ratio ($\nu=0.3$, see response to the following point 3), and the volumetric compaction observed under osmotic pressure loads (Fig. S10)

3. What is the Poisson's ratio of the hydrogel?

[Note: Similar response to Reviewer 4, point 1]

We measured the Poisson's ratio of the hydrogel by synthesizing hydrogel 'strings' in a glass capillary, and stretching the fluorescent strings under a microscope (details provided in Supplementary Material, Fig. S9, and Fig. 2b). The Poisson's ratio was experimentally determined to be 0.3 for strains less than 120%, which captures the operational range of MSG deformations observed in culture.

Figure S9. Measurement of Poisson's ratio of the fluorescently-labelled polyacrylamide hydrogel formulations used in this work. (a) Polyacrylamide hydrogel "strings" were fabricated within glass capillaries (internal diameter of 1.3-1.6 mm) that had been pre-treated to be hydrophobic. Following gelation, polyacrylamide hydrogel strings were released from the glass capillary and swelled for 24 hours before (b) stretching axially (scale bar = 1 mm) under a fluorescent dissecting microscope. The deformations in the transverse and axial directions were measured to compute the Poisson's ratio which (c) remains constant for strains up to 120% (data repeated in main manuscript Fig. 2b).

4. I. 178-179, Fig. 3e: *“At the MSC edge, tensions of up to +30 Pa were measured in the circumferential direction only.” The data in Fig. 3e show tensile stresses at the edge that are > 100 Pa.*

We apologize for this confusion. The manuscript has now been thoroughly checked for consistency with all our new simulations and analyses. Based on the updated simulations, the maximum tension observed is 52 Pa, in the circumferential direction only.

5. I. 238-241, Fig. 5e: *What is the stiffness gradient in the spheroid that is required for the stress distribution obtained from the FE model to be consistent with the experimental data? How is the stiffness gradient related to the stress gradient?*

Since the new analysis of the experimental data does not reveal statistically significant differences in stress between the intermediate and core regions (Fig. 5d, e), we cannot make a firm conclusion about quantitative stiffness patterns within the spheroid. We can only conclude that a stiffness gradient could result in the observed trend towards reduced forces, and that this is a reasonable assertion given the reduced density of cells in the central region of the spheroid (Fig. 6a). The models presented do not capture a stiffness gradient into the spheroid, but only tests the idea that a “step” difference in stiffness may dissipate forces caused by compaction from the spheroid edge.

[Redacted]

Reviewer #4 (Remarks to the Author):

1. The main novelty of the work is claimed to be replacing incompressible oil droplets (ref. #38) with compressible hydrogel particles that exhibit an isotropic linear elastic mechanical behaviour. While Poisson's ratio (ν) is the major factor controlling volumetric changes in such compressible materials; the authors fail to account for this quantity in the manuscript.

[Note: Similar question raised by Reviewer 3, point 3]

We measured the Poisson's ratio of the hydrogel by synthesizing hydrogel 'strings' in a glass capillary, and stretching the fluorescent strings under a microscope (details provided in Supplementary Material, and Fig. S9). The Poisson's ratio was experimentally determined to be 0.3, across a wide range of strains (data also presented in Fig. 2b).

Figure S9. Measurement of Poisson's ratio of the fluorescently-labelled polyacrylamide hydrogel formulations used in this work. (a) Polyacrylamide hydrogel "strings" were fabricated within glass capillaries (internal diameter of 1.3-1.6 mm) that had been pre-treated to be hydrophobic. Following gelation, polyacrylamide hydrogel strings were released from the glass capillary and swelled for 24 hours before (b) stretching axially (scale bar = 1 mm) under a fluorescent dissecting microscope. The deformations in the transverse and axial directions were measured to compute the Poisson's ratio which (c) remains constant for strains up to 120% (data repeated in main manuscript Fig. 2b).

2. In relation to this, in section “MSG mechanical characterization ...” on page 5, the shear modulus of $G=55 \text{ Pa}$ is mentioned while on pages 3 and 5 elastic modulus of $E \gg 100 \text{ Pa}$ is stated. Considering $G=E/[2(1+\nu)]$, where ν is the Poisson's ratio, implies $\nu \gg -0.1$ implying MSGs would dilate as they experience compressive stresses. Nevertheless, in Fig. S10, the results of three simulations with $\nu=0, 0.3$, and 0.499 were presented and it is mentioned in the caption that incompressible materials yield a singular matrix. At the end, it is unclear which value was adopted for Poisson's ratio ν (any of $0, 0.3$ or a negative value might have been considered).

The reviewer is absolutely correct, and we sincerely apologize for the confusion. We have thoroughly examined and corrected our manuscript to check for consistency in all the simulations, calculations and presented data, and in the presented figures and figure captions. To address these issues, we have:

- a) Experimentally measured the Poisson's ratio over a wide strain range, and used this value ($\nu = 0.3$) in all simulation data. (details in Supplementary Methods, Fig. S9 and, Fig. 2b)
- b) Measured the stiffness of MSGs directly using an osmotic pressure test, and found $G = 60 \text{ Pa}$ (details in Supplementary Methods, Fig. S10).
- c) These values were then converted to the two Lamé coefficients used to describe the compressible Neo-hookean hyperelastic material model employed in our updated simulations.
 First Lamé coefficient: $\lambda = \frac{2(1+\nu)\mu\nu}{(1+\nu)(1-2\nu)} = \frac{2(1+0.3) \times 60.018 \times 0.3}{(1+0.3)(1-2 \times 0.3)} = 90. \text{ Pa}$
 Second Lamé coefficient is just the shear modulus: $\mu = G = 60. \text{ Pa}$
- d) Finally, Fig. S12 (comparison of the effect of poisson ratio) has been updated with new simulation results, but still shows that incompressible materials produce non-deterministic solutions for radial and axial stress.

3. On page 3, the authors state: “their [oil droplets] incompressibility does not allow isotropic deformation”. However, it is unclear why only shear rheometry was employed to extract the shear modulus. Since calculation of Poisson's ratio is essential in this study; dilatational measurements in addition to shear rheometry can yield an estimation of both shear and volumetric mechanical properties and would be highly recommended.

[Note: similar response to Reviewer 3, point 3]

We agree, and as outlined in the response to the previous concern, we have measured the Poisson's ratio of the material directly, rather than rely on literature values. Our experimentally measured Poisson's ratio ($\nu = 0.3$) is consistent with the literature reports cited in the previous version of this manuscript.

4. Authors considered linear superposition to derive components of the stress tensor from measured components of the strain tensor. This principle is only valid for linear elastic materials. However, as depicted in Fig. 2 and Fig. S10 the stress-strain relationship is non-linear. This observed behaviour (for linear elastic materials) is normally referred to as geometrical nonlinearity (not material nonlinearity)

which arise from the spherical shape and occurrence of large strains within the linear elastic MSGs. Therefore, the overall behaviour of MSGs can be non-linear, and application of superposition principle would give errors in estimation of stress components. These issues need to be discussed in a rational and complete manner.

After developing the appropriate simulations to test the validity of linear superposition, we believe the reviewer is absolutely correct, and that **our previous approach of linear superposition is not valid**. Briefly (based on feedback from Reviewer 3, point 1), we developed strain-controlled models of MSG deformation to account for viscous flow of the surrounding tissue, using a hyperelastic neo-Hookean compressible material description to obtain realistic non-linear behaviour. Using these updated models, we computed the results of combinations of radial and circumferential strains, to determine stress state for several data points. We then compared these data points with similarly formulated models that use the linear superposition paradigm previously outlined. The results, particularly for compressive strains, were off by up to an order of magnitude. Hence, the reviewer is correct and linear superposition is not a valid approach, particularly for those MSGs under compression. We are extremely grateful for this insight, and have developed a new parametric computational approach to avoid linear superposition.

In our new simulations, finite element parametric sweeps for radial and axial strains in the MSG were varied. These results provide a calibration curve for axial and radial stress as a function of observed axial and radial strain. This information (Fig. 2e, f) was then used to interpolate stresses in each of the collected data points (results in Supplementary Table S2; Table S4).

Figure 2e, f (updated). (e, f) Axial (z) and radial (r) strains in an MSG are swept parametrically to determine the (e) axial (σ_z) or (f) radial (σ_r) stresses associated with each combination of strain. Using these simulations, observations of MSG deformation in the radial and axial directions can be used to determine the unique combination of stresses present at that location.

This improved analysis changes some of our key findings and results in this work. Although our analysis of tension has not changed significantly, compression values are much higher and more widely spread than previously predicted. Hence, compressive forces observed within a spheroid are far greater than previously reported, and are maintained by a relatively low tension in the outer ‘skin’ of the spheroid. Furthermore, differences in stress profiles in the intermediate and core regions of the spheroid are no longer as clear, and not statistically distinguishable (Fig. 5d, e).

5. In the abstract, “The Pascal-scale sensitivity of this technology can capture very small stresses ...” is mentioned. However, based on data shown large errors can exist in the presented calculations.

In light of the results mentioned in the response to the previous question, we have re-written the manuscript to accurately reflect our new findings, and conducted a thorough error analysis as follows:

5a. As shown in Fig. 2a, material parameters are not constant numbers due to errors in estimating these parameters.

We have now measured the mechanical stiffness of MSGs directly, and determined that they have an apparent shear modulus $G = 60$ Pa, and their values can vary by ± 3.5 Pa, which is within the measurement error of the mechanical characterization technique (± 7 Pa based on a 1 pixel measurement error in images). Since there is no correlation between individual MSG stiffness measurements and the size of the MSGs (Fig. 2c), it is reasonable to assume that MSGs produced within the same batch are of the same stiffness, and this variation hences affects **systemic accuracy** of the measurement technique. We analyzed our datasets assuming that MSG stiffness varies by 3.5 Pa, and determined that stress measurements are accurate within 6% (see Fig. S16). These errors have now been included in Figure 4 where they have been combined with measurement precision errors (outlined in the response to reviewer comment 5b).

Figure S16 (updated). Comparison of MSG errors associated with uncertainties in MSG modulus (accuracy) and strain measurement error (precision) in the (a) radial and (b) circumferential directions at day 2 of culture. Red data points represent tensional stress measurements, blue data points represent compressional stress measurements, and black data points represent stress measurements close to zero (-10 Pa to +10 Pa). Insets depict closer view of measured tensional stresses. Accuracy errors correspond to errors of 6% in stress readings, while precision errors were generated based on Monte Carlo simulations of error assuming a Gaussian normal distribution of values for repeated measurements of radial and circumferential bead dimensions. Both errors are combined and reported in the main manuscript figures (Figure 4). -----

5b. From Fig. 3c, it is unclear how accurately the periphery of the MSGs could be determined. Consequently, there would be an error in measuring its dimensions, and consequently in estimating strains.

Variations in determining the periphery of the MSG would affect **measurement precision** of the technique. Our images were collected using an sCMOS camera (2048 x 2048 pixels) and a 10x long-working distance objective, which produces an imaging resolution of 0.66 $\mu\text{m}/\text{pixel}$. Measurements of MSG dimension was determined by fitting ellipses to multiple points identified on the MSG edge, as outlined in Supplemental Methods. To determine the error that arises from this approach, we repeated this measurement three times on each MSG to determine the standard deviation of measurement error in the axial and radial MSG directions. While error propagation is relatively straightforward for linear systems, the non-linear models used here to relate deformation and stress cannot be propagated analytically. Hence, we developed a Monte Carlo simulation to determine the errors in stress values arising from errors in measuring MSG deformation. We randomly generated a normal Gaussian distribution of stress measurements for each MSG, and report the 95% confidence interval for each data point (Fig. S17 for representative sample distributions for a single data point).

Figure S17 (updated). Monte Carlo propagation of strain measurement uncertainty for the non-linear stress fit. Representative histograms of Monte Carlo estimates in stress measurement uncertainties for a single datapoint arising from errors in measurement of MSG deformation. Repeated measurement of MSG dimensions was used to estimate the error in analysis of MSG size along the axial and radial axes. Assuming a Gaussian normal distribution of measurements in both the radial and circumferential axes for each data point, 10,000 randomly generated deformation values were converted to stresses through the non-linear interpolation function described in Fig. 2e, f. (a, b) Representative datasets from (a) axial and (b) radial stress Monte Carlo statistical distributions for a single axial compression-radial tension MSG data point (-6.50 Pa in the axial direction; +40.46 Pa in the radial direction). Mean stress values (dashed line) and their respective 95% confidence intervals (green section) are obtained empirically from the randomly generated dataset around each point. Similar curves were generated for every datapoint analyzed, and the 95% confidence intervals for each point are plotted as estimates of error in Fig. S16. These errors are then combined with errors in systemic accuracy to determine the total measurement error, values reported in Figure 4, and in supplemental tables.

To combine the accuracy and precision errors, we report a linear combination of the errors in the main manuscript, as they both quantify an area of uncertainty in our measurement technique (Fig. 4), and acknowledge that the overall system of measurement is accurate within $\pm 10\%$.

Figure 4 (updated). Spatial mapping of cell-generated stresses within 3D fibroblast spheroid cultures. Internal tissue stresses in the (a) radial and (b) circumferential direction are measured at day 2 of culture. Low cell stresses are measured at the core, with increasingly compressive radial and circumferential stresses measured in the intermediate zone. At the very edge, tension is measured in the circumferential direction. No measurement in the radial direction is possible due to the free surface. Red data points represent tensional stress measurements, blue data points represent compressional stress measurements, and black data points represent stress measurements close to zero (-10 Pa to +10 Pa). Error bars indicate the 95% confidence interval for each data point, determined using a combination of systemic accuracy errors in defining MSG shear modulus, and Monte Carlo simulations of stress measurement arising from precision-related errors in identifying the MSG edge.

5c. Application of the linear superposition principle for non-linear materials would also give rise to errors. Furthermore, due to the non-linear geometry, the stress-strain curve is not symmetric with respect to the origin. Therefore, in this case there is a stress-induced anisotropy which was not clearly acknowledged.

As previously discussed, this considerable source of error arising from this analysis approach has now been replaced with parametric finite element analysis.

5d. Considering above sources of error, it is recommended that the authors perform error analysis and report the actual sensitivity of their mechano-sensors.

We agree and hope that the previous description of analyses satisfies this request.

6. Referring to Fig. 2, the finite element simulation is not well explained in the manuscript. For example, there is no information about the element type, mesh size, and whether effects of non-linearity (definition of large strains or large deformation) have been taken into account.

We apologize for the oversight. These simulations have now been completely updated in response to reviewer feedback, and hence the previous version has been removed from the manuscript. We have now provided a detailed description of the simulations currently used in the Supplemental Methods section, as follows:

Finite element analysis of microsphere deformation. Two-dimensional axisymmetric finite element models of the MSGs were developed in COMSOL v.5.3.1.201 (Comsol Inc.; Burlington, MA, USA) for the purpose of quantifying axial and radial stresses associated with experimentally-observed MSG deformation. Non-linear stress-strain behavior was captured by simulating the bead as a Neo-Hookean material with Lamé parameters calculated from experimentally-determined shear modulus and Poisson's ratio values (standard simulations performed with $\nu = 0.3$, $G = 60$ Pa, or Lamé 1 = 60 Pa, Lamé 2 = 90 Pa). MSG compression and expansion along the axial and radial directions were simulated with strain conditions applied to the MSG solid domain. The radial and axial stresses resulting from the simulated bead deformation were uniform through the MSG for both axial and radial stresses, and read directly from the simulations. A free quad mesh was used and optimized to ensure that the coefficient of variation was less than 1% (mesh element size was 2% spherical diameter; mesh element quality maintained above 0.8 for all strain combinations). A dual parametric sweep of axial and radial strains was performed to capture all possible bead deformation combinations. The stress-strain results were then assembled in MATLAB R2017b (The MathWorks, Inc.) into two surface/contours plots using a piecewise linear interpolation fit. Multicellular spheroid MSG deformation data was then processed using the interpolation fit to obtain the associated axial and radial stresses. Additional parametric sweeps were performed for the Poisson's ratio to demonstrate the effect of compressibility on the MSG's ability to read-out stresses ($\nu = 0, 0.3, 0.499$, results in Fig. S12), and for variations in the apparent shear modulus of MSGs to assess for MSG sensitivity. A deviation of ± 3.5 Pa in the shear modulus results in a $\pm 5.77\%$ error on the stress measurement (Fig. S16).

7. Figures 3d and 3e show that stresses as high as ± 100 Pa have been calculated. However, the magnitude of stress in the stress-strain curves represented in Figures 2c and 2d (or Figure S10) was not extended to such high values. Therefore, estimations of ± 100 Pa are questionable and in the results of FEM simulations the stress axis needs to be extended so it contains the ± 100 Pa range.

We agree with this concern, and our new parametric simulations have been developed to span the entire range of observed strains in the MSGs.

8a. In relation to this comment and with regard to Fig. 3d and Fig. 3e, for a MSG located near the edges of the spheroid (distance to edge is a few micrometres) the radial and circumferential stresses take values near -100 Pa and $+100$ Pa, respectively. In other words, with that stress and FEM data, we expect that the MSG changes from a spherical to ellipsoidal shape due to a compressive stress of 100 Pa acting along the minor axis of the ellipsoid and a tensile stress of similar magnitude acting along its major axis. Conservatively, we ignore the circumferential stress and assume that only radial

compressive stress is being applied to the ellipsoid. By extrapolating stress-strain curves in Fig. 2d or Fig. S10a or S10b (the $\Delta r/r_0$ - σ curves), at $\sigma=100$ Pa, the strain will be more than ~ 1 , which is not possible. In other words, accepting Fig. 2 or Fig. S10 as a valid constitutive law is inconsistent with observed/calculated stresses as high as 100 Pa.

We agree, and hope that our updated simulations results that do not rely on linear superposition, reflect a thoroughly characterized material dataset, and span the entire range of observed MSG deformations address these concerns. As a minor point, we would also like to mention that this analysis requires knowledge of the zero-stress state of the MSG, which was not present in the images, and which would considerably affect the reviewers' analysis.

8b. Furthermore, Fig. 3c shows a slightly deformed ellipsoid (egg shape) near the spheroid edge, which seems inconsistent with the stress-strain relationship presented; it should be much more elongated in that region.

The images presented in Fig. 3c were originally intended to be illustrative. We have now included data showing MSGs from select positions that should accurately capture the analysis presented here, and provide information on the actual location of the MSGs within the spheroids.

Figure 3e (updated). Embedded fluorescent sensors deform within the spheroid, with circumferential, radial, or no orientation (scale bar = 50 μ m), based on position within the spheroid

9. Why are the results presented in Figure S11 (showing non-linear stress-strain curves) different from those in Fig. S10 (showing linear stress-strain)? Furthermore, what is the difference between Fig. S11a and S11b?

The explanation for these discrepancies is now irrelevant to the manuscript because these poorly-executed figures (Fig. S10; Fig. S11) have been removed based on a substantially improved understanding of this system, and a realization that stiffness mismatch is irrelevant when the tissue is considered to exhibit viscous flow around the MSG (see response to Reviewer 3, point 1).

10. *Fig. 3d shows 12 data points while Fig. 3e 17. The authors should mention clearly how many MSGs and how many spheroids were used to draw Figs. 3d and 3e.*

We have now provided all our raw data in a supplemental table (Table S1-4). We conducted all our experiments using one MSG per spheroid, or 35 spheroids for our largest dataset. These numbers have now been included in the Methods section of the manuscript, as well as in the captions of the relevant figures.

11. *Authors should use the results of more tests to fill the gap between 250 μm and 500 μm in Figs. 3d and 3e. Moreover, referring to Fig. 3e at close distances to the edge (distance $\gg 0$) only one data point exists (with $\sigma\theta \gg 110 \text{ Pa}$). More data are needed close to the edge.*

[Note: similar response to Reviewer 1, point 1; and Reviewer 2, point 4b]

Re: Measurements at the surface

We have obtained more data points close to the edge of the spheroid, and consistently show that the outer 30 microns of the spheroid are under tension (updated Fig. 4b). Finding MSGs in this region is challenging, because the surface layer under tension is only a few cells thick, and if the MSG “breaches” the surface it is quickly ejected. We do however consistently observe this tension effect in all our experiments

Re: Measurements at the core

We agree that the single point is not sufficient. To add more data, we attempted additional experiments, and although we improved our sample size (Fig. 3f; Fig. 4), we could not image MSGs deeper than 300 μm below the MCS surface, likely due to the optical density of the spheroids. We attempted two-photon confocal microscopy but were unable to penetrate further without unacceptable signal losses. Hence, we removed the single data point as it could not be repeated at this depth. Our additional data indicates that there is a small decrease in the average stress in the core region compared to the intermediate zone, but this difference is not statistically significant ($p = 0.24$ by two-tailed ANOVA), likely due to the high spread in measurement values. Hence, this does not allow us to conclude that the stress plateaus. We have therefore restated this observation, incorporated our new data and analysis, and explicitly acknowledge the limitation of this fluorescent optical imaging modality in the discussion section of the manuscript. The relevant updated datasets and analysis are reproduced in this response.

Selected panels from Figure 4 and Figure 5 (updated). Spatial mapping of cell-generated stresses within 3D fibroblast spheroid cultures, demonstrating additional data points collected within the spheroid. Internal tissue stresses in the (a) radial and (b) circumferential direction are measured at days 2 of culture (Figure 4). Using the updated analysis models developed in response to reviewer comments, compressive stresses in the radial and circumferential directions as large as 1.3 kPa are measured within the spheroid, while (inset) tensional stresses are measured in the circumferential direction at the outer surface of the spheroid. In Figure 5 (d, e) stress measurements are grouped based on location for statistical comparison. * indicates $p < 0.05$ (one-way ANOVA with Tukey post-hoc comparisons).

12. The authors used an “Olympus, IX73” microscope which is presumably an epifluorescence microscope. However, it would be much more informative if the authors would use a confocal microscope and present 3D reconstructed images of an ellipsoidal MSG and a stress-free spherical MSG obtained after lysing the spheroid. Such images could be added to Fig. 3. Furthermore, the authors indirectly assumed, probably due to symmetry of the spheroid, that the two circumferential components of stress are identical. This assumption needs to be verified and it is essential to show a 3D image of the ellipsoid with two axes of similar length.

The reviewer is correct and we did assume that the symmetry of the spheroid leads to axisymmetric deformation of the MSG. To confirm this, we conducted the requested high-resolution confocal scans. While epifluorescent imaging with a bright LED light source was sufficient to obtain 2D-projection images of MSGs up to 300 μm below the surface of the spheroid, obtaining high-resolution 3D confocal images deep within the tissue was not possible because of unacceptably high out-of-plane signals, even using two-photon confocal microscopy. We were hence able to collect the requested 3D confocal

images of MSGs close to the spheroid surface (Fig. S14), and our reconstructed images do confirm symmetric deformation where the two circumferential stress components are similar.

To address this important issue in the manuscript, we have:

- 1) explicitly stated the assumption that stresses within the spheroid tissue will present spherical symmetry (“MSG mechanical characterization and analysis of deformation” section):

“Although a full 3D finite element analysis for individual MSGs is possible, we took advantage of the symmetry inherent in our intended application to simplify the models required. Hence, we assumed uniform tensile and compressive loads along the MSG axial (z-direction), and radial (r- direction) axes (Fig. 2d),...”

- 2) explicitly state that this analysis is a simplification meant to capture overall MSG deformation in response to “tissue-scale” stresses, and does not account for sub-MSG and highly localized stresses generated by individual cells, which would require high-resolution imaging and analysis of each MSG. (Discussion section), as follows:

Finally, the analysis technique developed here considers only ellipsoidal geometries for MSG deformation, and explicitly assumes that stresses within an MSG are uniform and have at least one axis of symmetry. While this is justified for analysis of supra-cellular stresses in MCS cultures, this approach does not capture sub-cellular stresses that must be present, and would not be sufficient to analyze deformation in a non-symmetric tissue. Fortunately, computational approaches to model mechanical stresses in each individual MSG can be readily developed⁴¹.

- 3) presented the requested confocal images to confirm this assumption (Fig. S14).

Figure S14 (updated). Reconstructed confocal images of MSGs embedded at the periphery of the MSG confirms symmetric deformation of the MSGs within spheroid cultures. (a) Schematic representation of the MSG location within the spheroids, and (b) reconstructed confocal images of MSGs close to the surface show deformations as expected based on spherical symmetry (scale bar = 25 μm), (c) The ‘pancake’-like morphologies adopted demonstrate two main axes of uniform deformation (radial and circumferential), arising from compressive radial stress and tensional circumferential stress.

[Redacted]

14. *Forces acting inside the spheroid have a dynamic nature and their magnitude and direction change as a function of time and during spheroid formation. The authors measurements were only carried out at a particular time on the second day of culture. However, it would be interesting to measure the variation of radial and circumferential stresses at a specific location versus time (for instance, at day 1, 2 and 3). This would enable the authors to fairly assess the power of the proposed method to assess the “stresses arising during multicellular spheroid aggregation” as mentioned in the abstract.*

We agree, and have carried out the suggested experiments. The results (now included in the manuscript as panels a, b, e, and f in Fig. 4) clearly demonstrate increasing stress levels over time, and decreasing stress levels when the tension-generating machinery of the cell is disrupted. A discussion on these results has also been included in the discussion section of the manuscript.

Figure 4a, b (updated). Spatial mapping of cell-generated stresses within 3D fibroblast spheroid cultures. (a-d) Internal tissue stresses in the (a) radial and (b) circumferential direction are measured at days 0 ($n = 9$), 1 ($n = 17$), and 2 ($n = 35$) of culture. Low cell stresses are measured at the core, with increasingly compressive radial and circumferential stresses measured in the intermediate zone. At the very edge, tension is measured in the circumferential direction. Red data points represent tensional stress measurements, blue data points represent compressional stress measurements, and black data points represent stress measurements close to zero (-10 Pa to +10 Pa). Error bars indicate the 95% confidence interval for each data point, determined using a combination of systemic accuracy errors in defining MSG shear modulus, and Monte Carlo simulations of stress measurement arising from precision-related errors in identifying the MSG edge.

Figure 4e, f (updated). The average stresses measured throughout the MCS in the (e) radial and (f) circumferential directions increased significantly from day 0 to days 1 and 2, and was significantly reduced by inhibition of actomyosin contractility. * indicates $p < 0.05$ (one-way ANOVA with Tukey post-hoc pairwise comparisons).

15. The authors neglected the possible effects of time on the mechanical behaviour of the MSGs. The employed PA gels are known to be elastic, however for the low modulus material used in this study, the authors should provide evidence for the assumed constant elastic mechanical behaviour (non-plastic deformations) over the 3 days of experiment. This could be simply done through lysing the spheroid

and showing the images of MSGs (preferably those located at the edges where stress/ strain is high) at day 3.

This is an excellent suggestion, and we have confirmed that MSG sizes return to their original shape after the spheroid is lysed (see Fig. 4; stresses observed at day 0 are ~ 0 Pa, reflecting no changes in dimension at the beginning and end of the experimental culture time). In addition, we have also applied our osmotic pressure measurement technique to determine the mechanical properties of individual MSGs before culture and after spheroid lysis. This analysis demonstrate no change in mechanical properties after the MSGs are incorporated into the spheroid (Fig. S10g). These results have been discussed in the manuscript (MSG mechanical characterization section), and taken together strongly demonstrates that MSG properties remain constant during culture.

Figure S10g. MSG mechanical properties remain constant after collagen coating and release from MCS culture. A long-chain dextran solution (500 kDa) was used to exert an osmotic pressure on polyacrylamide hydrogels. The dextran molecules are too large to enter the polyacrylamide pores, and are therefore excluded from the hydrogel. No significant differences were found between coated MSGs (control) and MSGs that had been removed from spheroids after two days of culture by detergent-based extraction (released), demonstrating that MSG properties remain constant even after embedding within the tissue of interest ($n = 16-19$, $p = 0.837$). All data reported as mean standard deviation. NS indicates no significant differences (one-way ANOVA with Tukey post-hoc pairwise comparisons).

Minor:

- On page 6, the authors mentioned: “However, when the tissue can plastically reorganize and flow around the MSG, as in the MCS culture models presented here, stresses will be redistributed to minimize mismatch-driven variations”. Authors should provide more evidence to support this statement.

We have now provided additional descriptions of this phenomenon as well as references to relevant literature that supports considering cell aggregates as a viscous fluid material over the appropriate time scales. The text inserted in the discussion section of the manuscript is as follows:

Second, the viscous nature of the biological tissue being studied must be carefully considered. Most biological tissues exhibit some form of viscoelastic behaviour, and behave as elastic solids on short timescales and as fluids on long time scales⁵⁶⁻⁵⁸. If the tissue behaves like an elastic solid, then the MSG within the tissue can be considered as an inclusion within an elastic matrix. In such cases, the modulus of the surrounding tissue and the contact boundary conditions at the MSG-spheroid interface plays a critical role in deformation of the MSG⁵⁹. Hence, the presented analysis applies only if the tissue exhibits fluid behaviour to dissipate inclusion-generated stresses. For example, measurements taken seconds after cells in a collagen matrix are stimulated to contract by addition of a chemical factor would not allow the cells to flow and would therefore not provide accurate stress

measurements. In the current case of a MCS in which cells form transient adhesions and can migrate over each other, the time constant of solid-liquid transitions has previously been established to be on the order of minutes⁶⁰. Although tissue components such as secreted extracellular matrix proteins could slow cell movement, we confirmed that ECM in our spheroids is not in mature fibrillar form (Fig. S13b) and is hence unlikely to stabilize the fluid behaviour beyond this time frame. Hence, measurements taken in spheroids at 24 hour intervals will not be affected by this factor.

- On page 9, line 13, a sentence started with a lower-case letter (*finite element models...*).
- On page 13, last line, "Dolega et al" the point is missing (*Dolega et al.*).

Thank you, we have addressed these issues.

Reviewers' Comments:

Reviewer #1:

Remarks to the Author:

In my opinion, the authors have done an excellent job and significantly improved their original study and manuscript.

I have only a few minor questions and suggestions left.

The Blebbistatin experiments could be described more precisely. Are the cells remaining in the same medium and drug the entire time (3 days)? Is the drug still active after that long? One easy way to check is to wash out the drug and see if the aggregates reach the compaction state of the control aggregate after wash.

I read that the authors have used the racemic mixture of Blebbistatin instead of the purified active enantiomere (the (-) one). That is not ideal as only half of the compound is active and Blebbistatin suffers from solubility problems (especially at concentrations as high as 50 μM). Also, the inactive enantiomere (the (+) one) constitutes an excellent control for the drug. What did the authors use as control? I could not find this information.

In Fig 3e, the dashed line around the MSGs is not really helping in appreciating their shape without any bias. Can the authors remove it?

There seems to be some problem with the labels in Fig 5f and 5h.

To analyze the alignment of cells with the circumferential stress, the authors could look into the shape of the whole cell instead of the nucleus only. Indeed, the nucleus will follow the deformation of the cell only if the deformation is large and therefore, measuring the nuclear deformation is not very sensitive. This could solve the disagreement between the authors impression and quantification attempts.

Reviewer #2:

Remarks to the Author:

The authors have satisfactorily addressed my major concerns in their revised manuscript. I am supportive of publication of the manuscript.

Reviewer #3:

Remarks to the Author:

General Comments:

The authors responded to all my comments and revised the text accordingly. I highly appreciate their effort. Most of the responses to my comments and the corresponding revisions are satisfactory except for the comments regarding the inclusion of MSG in the tissue for the following reason (l. 328-344).

The authors acknowledged that over short time scales (less than a minute) tissue behaves like an elastic solid and hence "the MSG within the tissue can be considered as an inclusion within an elastic matrix. In such cases, the modulus of the surrounding tissue and the contact boundary conditions at the MSG-spheroid interface plays a critical role in deformation of the MSG." They further speculated that over longer time scale (order of minutes) the tissue behaves as a viscous fluid, "in which the modulus of the biological cell/tissue, and the nature of the boundary between the tissue and the underlying polyacrylamide substrate is not relevant to the analysis." Strictly speaking, the latter argument would hold only if the tissue exhibits a Newtonian viscous behavior.

In my knowledge, majority of soft tissues, including lungs, cartilage, skin, muscles, etc., do not exhibit a Newtonian-like behavior within physiological range of time scales. Rheological measurements on soft tissues indicate that over a wide range of time scales (many orders of magnitude) they follow a power-law rheology. (For example, lung parenchyma exhibits a power-law relaxation over time scales on the order of human life time.) This type of behavior is fluid-like, but it is a non-Newtonian fluid behavior. Power-law rheology is characterized by finite elastic (storage) and viscous (loss) modulus. Thus, I would argue that even over time scales of several minutes to hours (or longer) the surrounding tissue has a finite elastic modulus and hence contact boundary conditions at the MSG-spheroid interface should play a role in deformation of the MSG. If so, then the accuracy of stress estimates in soft biological tissues by the proposed method is still questionable.

Minor comments:

I. 255: Please define "H & E strains".

Reviewer #4:

Remarks to the Author:

The revised version of the manuscript is significantly improved. We have only two remaining minor comments:

Figure 1 b shows a very asymmetric inhomogeneous deformation of a microsphere which is misleading since in this study only symmetric radial and axial deformations are considered. This needs to be corrected and very clearly stated at the beginning of the manuscript.

In Figures 2e and f, labels needs to be adjusted.

We thank the reviewers for their time, effort and insight in re-examining our revised manuscript. We were extremely pleased with the quality of the reviewer feedback, gratified to see that our efforts were appreciated, and are happy to address these remaining issues. We have conducted additional experiments, revised our discussion and language,, and provided additional supporting evidence from the literature to clarify these remaining concerns, as detailed in the following point-by-point response.

Reviewer #1 (Remarks to the Author):

1a. *The Blebbistatin experiments could be described more precisely. Are the cells remaining in the same medium and drug the entire time (3 days)? Is the drug still active after that long? One easy way to check is to wash out the drug and see if the aggregates reach the compaction state of the control aggregate after wash.*

We have now described the original experiments more precisely, and conducted the suggested study. As expected, we confirm that washing out the blebbistatin after 24 hours allows the spheroids to resume contraction and achieve their baseline diameters (now included as Figure S20 in the manuscript). This confirms the activity of the drug for the duration of the culture as previously, the addition of blebbistatin to the culture medium impeded spheroid contraction within the first 24 hours, and this behaviour was sustained over 3 days of culture.

b. *I read that the authors have used the racemic mixture of Blebbistatin instead of the purified active enantiomere (the (-) one). That is not ideal as only half of the compound is active and Blebbistatin suffers from solubility problems (especially at concentrations as high as 50 μ M).*

We agree that using 50 μ M of the racemic mixture may not be bioequivalent to using 50 μ M of purified enantiomer, due to issues with solubility and activity. However, since we observe the expected reduction in MCS compaction (Fig. 3b), we can conclude that a 50 μ M nominal concentration of racemic blebbistatin is sufficient to reduce cytoskeletal contractile activity as intended in this experiment. This is supported by our observations of reduced MSG deformation (Fig. S15, compared to Fig. 3f), and reduced measured stresses (Fig. 4), all of which is consistent with reduced cytoskeletal contractility.

c. *Also, the inactive enantiomere (the (+) one) constitutes an excellent control for the drug. What did the authors use as control? I could not find this information.*

Our initial experiment had set up the “0 μ M blebbistatin” case to be the control. Following this comment, we realized that we did not control for the vehicle used to solubilize blebbistatin (DMSO). We have now confirmed that the DMSO vehicle does not alter MSG compaction (included in new Fig. S20).

To address these points in the manuscript, we have:

- 1) Clarified that 50 μ M of blebbistatin is the nominal concentration used (Supplemental Methods)
- 2) Provided the following text in supplemental methods:

“To inhibit contractility, MCS cultures were treated by adding blebbistatin (stock solutions prepared in DMSO) to the culture media immediately after seeding the cells. MCS treated with 50 μ M blebbistatin (nominal concentration; Sigma-Aldrich, 203390) did not compact to the same extent as control cultures (Fig. 3B), or those

treated with only the DMSO vehicle (Fig S20). To confirm that blebbistatin did not permanently damage the contractile apparatus of the cells, it was washed out of culture, after which the MCS contracted to their baseline sizes (Fig S20).”

3) Included the following new data as supplemental figure S20.

Figure S20. Confirmation of blebbistatin activity through quantification of spheroid size. Compaction is significantly reduced in MCS when actomyosin contractility is inhibited with blebbistatin (day 1), but is regained when the drug is washed out (day 2). Data reported as mean \pm standard deviation, $n = 11$, * indicates $p < 0.01$ (Student's t-test).

2. In Fig 3e, the dashed line around the MSGs is not really helping in appreciating their shape without any bias. Can the authors remove it?

We have removed the dashed line in Fig. 3e, and uniformly adjusted the image contrast to clearly see the MSG shape.

3. There seems to be some problem with the labels in Fig 5f and 5h.

Thank you, we have corrected this.

4. To analyze the alignment of cells with the circumferential stress, the authors could look into the shape of the whole cell instead of the nucleus only. Indeed, the nucleus will follow the deformation of the cell only if the deformation is large and therefore, measuring the nuclear deformation is not very sensitive. This could solve the disagreement between the authors impression and quantification attempts.

Thank you for the suggestion. We have re-analyzed our H&E-stained tissue sections to quantify cell size rather than nuclear size. We do observe a small but statistically significant increase in cell elongation when comparing the edge to the core regions, consistent with visual inspection of the image. This new analysis has been included as an additional panel in Supplementary Figure S13, and we have cited this finding in the main text under “Observed stress patterns correlate with biomarkers of mechanical activity”.

Figure S13 (new panel f). Characterization of cell elongation in H&E stained spheroid sections indicate significant differences in cell elongation between the edge and core sections. Data reported as mean standard deviation, $n = 30$ over 3 spheroids, * indicates $p < 0.001$ (Student's t-test).

Reviewer #2 (Remarks to the Author):

The authors have satisfactorily addressed my major concerns in their revised manuscript. I am supportive of publication of the manuscript.

Thank you.

Reviewer #3 (Remarks to the Author):

1. *The authors acknowledged that over short time scales (less than a minute) tissue behaves like an elastic solid and hence “the MSG within the tissue can be considered as an inclusion within an elastic matrix. In such cases, the modulus of the surrounding tissue and the contact boundary conditions at the MSG-spheroid interface plays a critical role in deformation of the MSG.” They further speculated that over longer time scale (order of minutes) the tissue behaves as a viscous fluid, “in which the modulus of the biological cell/tissue, and the nature of the boundary between the tissue and the underlying polyacrylamide substrate is not relevant to the analysis.”*

Strictly speaking, the latter argument would hold only if the tissue exhibits a Newtonian viscous behavior. In my knowledge, majority of soft tissues, including lungs, cartilage, skin, muscles, etc., do not exhibit a Newtonian-like behavior within physiological range of time scales. Rheological measurements on soft tissues indicate that over a wide range of time scales (many orders of magnitude) they follow a power-law rheology. (For example, lung parenchyma exhibits a power-law relaxation over time scales on the order of human life time.) This type of behavior is fluid-like, but it is a non-Newtonian fluid behavior. Power-law rheology is characterized by finite elastic (storage) and viscous (loss) modulus. Thus, I would argue that even over time scales of several minutes to hours (or longer) the surrounding tissue has a finite elastic modulus and hence contact boundary conditions at the MSG-spheroid interface should play a role in deformation of the MSG. If so, then the accuracy of stress estimates in soft biological tissues by the proposed method is still questionable.

The reviewer is correct that the nonlinear behaviour of tissues over the experimental time scales could affect the accuracy of stress estimates using this method. We were wrong to state that “the nature of the boundary... is not relevant to the analysis”, and would like to withdraw this statement. We agree

that this analysis cannot be used without carefully considering the power-law rheology of the biological tissue being studied, and have updated the manuscript to more clearly reflect this limitation. We do however maintain that for the specific spheroids studied in this work, neglecting the nonlinear characteristics of long-term fluid-like behaviour has only a minimal impact on our estimates of stress.

To clarify our reasoning, we agree that the spheroid does maintain a finite elastic modulus over all time scales. Our data demonstrates stress variations within the spheroid, which would not be possible in a purely viscous fluid. Instead, spheroids are aggregates of contractile and adhesive particles which stick to each other under compressive or tensile stress; and viscously deform and migrate over each other to dissipate shear stress. Nonlinear characteristics such as a finite yield shear stress for aggregated cells under shear loading is established to be extremely small: load-bearing adhesions between cells support stresses of ~ 2 Pa across the adhesive structure (Grasshof et al., Nature 2010; PMID 20613844), and fibroblasts similar to those used in this work migrate in response to shear stresses as low as 2 Pa (Garanich et al., 2007, PMID 17308005). Other cell types migrate at even lower shear stresses (Ando et al., 1987, PMID 3561268). We also carefully verified that no load-bearing ECM components are present to stabilize our recently-aggregated spheroids, suggesting that any yield shear stresses between cells in our spheroids will be similarly low.

This understanding of shear-stress dissipating behaviour within spheroids has been well-established in the literature, through several seminal works outlined in our previous response. In addition to those studies, Yu et al., Biophys J. 2018 (PMID: 29874619) recently demonstrated that multicellular spheroid rearrangement dissipates shear stress within hours, and these timescales are universally conserved across many cell lines, regardless of broad variation in other physical properties. Hence, while we agree with and appreciate the reviewers' concerns regarding broad generality of this analytical approach, we believe that the accuracy of this method for this specific application over the 24-hour experimental sampling times studied here is reasonable.

To capture this important discussion, our misleading description that "most biological tissues... behave as elastic solids on short timescales and as fluids on long time scales" has been removed, and the discussion on limitations of the technology (Discussion, paragraph 4) has been rewritten, as follows:

Second, the remodelling capacity of the biological tissue being studied must be considered carefully prior to designing these experiments. The presented approach is only valid if the tissue remodels sufficiently to dissipate local shear stresses that arise around the MSG due to stiffness mismatches between the sensor and the tissue at the contact interface [59]. For example, if cells in a collagen matrix are stimulated to rapidly contract, the ultrasoft MSG will act as a void in the tissue and redistribute stresses around it, providing an inaccurate measurement of local stress. However, if these local shear stresses are dissipated through viscous remodelling of the tissue, deformation of sufficiently small MSGs will accurately reflect local stresses. Hence, the accuracy of this technique for a given sampling time depends on the power-law rheology of the tissue of interest, which may vary considerably from tissue to tissue [56–58]. Large multicellular spheroid aggregates are well-established to exhibit viscous deformation in response to small shear stresses within minutes [60-61], and since ECM in our spheroids is not expressed in mature load-bearing form (Fig. S13b), they are unlikely to provide additional resistance to local shear stresses over the 24 hour time points studied here.

Minor comments:

2. l. 255: Please define “H & E strains”.

H&E (Haemotoxylin and Eosin) stains are used to label tissue sections to image the nuclei (stained purple by haemotoxylin) and proteins in cytoplasm and the extracellular matrix (stained pink by eosin). We have now included this additional information in the main manuscript text.

Reviewer #4 (Remarks to the Author):

The revised version of the manuscript is significantly improved. We have only two remaining minor comments:

1. Figure 1 b shows a very asymmetric inhomogeneous deformation of a microsphere which is misleading since in this study only symmetric radial and axial deformations are considered. This needs to be corrected and very clearly stated at the beginning of the manuscript.

We agree, and have replaced this panel in Figure 1 with only the symmetric deformation that was previously included in Figure 2. We also specified in the introduction that we “quantitatively mapped highly localized and symmetric radial and circumferential stresses within the tissue”.

2. In Figures 2e and f, labels needs to be adjusted.

Thank you, we have adjusted the Figure 2 panel layout, increased the image size, and increased the text label sizes to improve readability.

Reviewers' Comments:

Reviewer #1:

Remarks to the Author:

Label is missing in Fig 5F.

Reviewer #3:

Remarks to the Author:

The authors satisfactorily addressed my concerns. I have no additional comments.